# Cross-Modal Generative Augmentation for Multimodal Biological Classification

**Hyunwoo Yoo**                                                              *hty23@drexel.edu*
*Department of Electrical and Computer Engineering*
*Drexel University*

**Efstathia Soufleri**                                                    *e.soufleri@athenarc.gr*
*Archimedes, Athena Research Center*

**Deepak Ravikumar**                                                     *dravikum@purdue.edu*
*Elmore Family School of Electrical and Computer Engineering*
*Purdue University*

**Gail L. Rosen**                                                            *glr26@drexel.edu*
*Department of Electrical and Computer Engineering*
*Drexel University*

**Reviewed on OpenReview:** *https://openreview.net/forum?id=bowYeHa8dn*

## Abstract

Recent advances in vision-language models have enabled cross-modal generation between text and images, achieving remarkable progress in general-domain understanding. However, their potential in scientific and biological applications remains largely unexplored, where datasets often couple complex visual observations with structured metadata or textual descriptors. We propose a cross-modal generative framework that supports direction-agnostic generation (image-to-text or text-to-image) depending on modality availability to enrich multimodal biological classification. Our framework integrates generative augmentation and multimodal alignment to provide complementary augmentation for visual and textual representations, enabling the synthesis of complementary modality data that may otherwise be unavailable in biological datasets. Experimental results on the HAM10000 and EMPO500 datasets demonstrate improvements across multiple evaluation metrics across diverse biological datasets over baseline models. The proposed framework is model-agnostic and compatible with open-weight alternatives, paving the way for biologically grounded multimodal generation and analysis.

## 1 Introduction

Large vision-language models (VLMs) have recently shown impressive capabilities in cross-modal generation, enabling the synthesis of semantically coherent text and images from multimodal inputs (Soufleri & Ananiadou, 2025; Koh et al., 2023; Chow et al., 2024). While these advances have transformed visual understanding and creative tasks, their potential in scientific and biological domains remains underexplored (Zhang et al., 2024). Biological and environmental datasets often couple complex visual patterns with structured metadata or descriptive annotations, for example, linking visual findings in dermatology to diagnostic attributes or connecting environmental measurements to contextual descriptions. In many biological datasets, however, one modality is often incomplete, noisy, or entirely missing due to limited annotations or acquisition constraints. Generative modeling therefore provides a practical mechanism for synthesizing complementary modalities that would otherwise be unavailable, enabling richer multimodal learning without requiring additional data collection. Bridging these modalities through generative modeling offers a promising path to enhance data efficiency (Du et al., 2022) and interpretability in biological classification.

However, existing multimodal approaches often focus on a single generative direction (Shi et al., 2024; Liang et al., 2024), employing either image-to-text or text-to-image generation in isolation. Such one-way mappings limit the ability to leverage complementary information across modalities and prevent the exchange of semantic knowledge that could otherwise reinforce both representations. Moreover, biological datasets are generally small, heterogeneous, and domain-specific, which amplifies the difficulty of cross-modal alignment and augmentation (Nam & Jang, 2024).

To address these challenges, we propose a cross-modal generative augmentation framework that supports direction-agnostic generation, where either image-to-text or text-to-image generation is applied depending on modality availability to synthesize complementary modality data for multimodal biological classification. In the HAM10000 dermatology dataset (Tschandl et al., 2018), the framework uses image-to-text generation to produce EHR (Electronic Health Record)-like diagnostic narratives from dermoscopic images and metadata. These generated descriptions are fused with visual embeddings through a lightweight vision–metadata fusion module, improving interpretability and classification accuracy. Conversely, in the EMPO500 microbiome dataset, text-to-image generation produces synthetic environmental imagery conditioned on tabular metadata, expanding the visual space for downstream training. Both directions employ generative augmentation as auxiliary components, while the overall design remains model-agnostic, allowing compatibility with both closed and open-weight generative models.

**Independent Cross-Modal Generative Augmentation.**

Our framework employs two independent generative augmentation pathways: image-to-text and text-to-image. These generators synthesize complementary modality data that are incorporated into a shared multimodal classifier.

Image-to-text generation expands semantic metadata grounded in visual evidence, while text-to-image generation expands the visual feature space conditioned on structured context. Rather than enforcing direct reconstruction or cycle-consistency objectives, the two pathways operate independently and interact through the shared multimodal classifier.

Extensive experiments demonstrate that our approach achieves competitive performance against strong unimodal and unidirectional baselines, improving both generalization and multimodal coherence across datasets. By coupling generative augmentation with representation-level fusion, the proposed framework provides a practical approach for biologically grounded multimodal analysis.

| Model / Framework | Visual + Text Fusion | Text→Image Generation | Image→Text Generation |
|---|:---:|:---:|:---:|
| ALBEF (Adebiyi et al., 2024) | ✓ | ✗ | ✗ |
| BiomedCLIP (Zhang et al., 2025) | ✓ | ✗ | ✓ |
| BackModality (Li et al., 2023) | ✗ | ✓ | ✓ |
| Multimodal Stress Detection (Soufleri & Ananiadou, 2025) | ✓ | ✓ | ✗ |
| TaxaBind (Sastry et al., 2025) | ✓ | ✗ | ✗ |
| **Our Work (Cross-Modal Gen.)** | ✓ | ✓ | ✓ |

Table 1: Comparison of multimodal and generative models. A checkmark (✓) indicates support for the corresponding capability. This table summarizes capability coverage across representative models, without implying novelty or priority of invention.

Our main contributions are as follows:

- We introduce a cross-modal generative augmentation framework that supports either image-to-text or text-to-image generation depending on the dataset to support multimodal biological classification.

- The framework enables the synthesis of complementary modality data that may otherwise be unavailable in biological datasets.

- We design a fusion strategy that combines generative augmentation and multimodal alignment, allowing textual and visual modalities to provide complementary information.

- We empirically validate the framework on two heterogeneous biological datasets, demonstrating consistent and competitive gains in accuracy, interpretability, and generalization.

## 2 Related Work

**Multimodal Learning in Biological and Biomedical Domains**   Recent multimodal learning approaches have demonstrated that integrating visual and textual modalities can improve performance in biomedical and ecological tasks (Zhang et al., 2025; Sun et al., 2023; Thapa et al., 2024). In dermatology, multimodal frameworks such as ALBEF-based systems (Adebiyi et al., 2024) have combined dermoscopic images with patient metadata (e.g., age, sex, lesion location) to achieve high accuracy on HAM10000 and ISIC datasets, showing the effectiveness of vision–metadata fusion for diagnosis. Similarly, multimodal encoders (Amar et al., 2025; Sastry et al., 2025) have been applied to microbiome and ecological data, where textual or tabular metadata complements visual information to enhance classification robustness under data imbalance (Yoo & Rosen, 2025a;b). However, these studies largely rely on fixed visual inputs and static text encoders, limiting their adaptability when one modality is scarce or missing.

**Generative Models for Multimodal Data Augmentation**   Generative modeling has recently emerged as a promising solution to bridge information gaps between modalities (Li et al., 2023; Xiao et al., 2023; Wu et al., 2024). Vision–language models such as diffusion-based text-to-image generators and large language models for captioning have shown potential in synthesizing biologically meaningful representations. Prior work has explored modality transformation for augmentation, for example Back-Modality (Li et al., 2023), as well as bidirectional multimodal learning for alignment or knowledge transfer in retrieval and contrastive settings (Wang et al., 2025; Zareapoor et al., 2025). Previous studies have shown that fusing real and synthesized images guided by textual context enhances multimodal classification performance in social and affective computing (Soufleri & Ananiadou, 2025). In biological settings, text-guided synthetic imagery has been used to augment rare classes in datasets like HAM10000 and EMPO collections, leading to improved generalization without manual data expansion. However, existing approaches differ in their objectives and deployment strategies, often focusing on either a fixed generative direction or tightly coupled bidirectional training, rather than flexible, dataset-dependent augmentation.

**Cross-Modal Generative Augmentation Framework**   Cross-modal generation and modality transformation have been explored in prior work, including bidirectional approaches for augmentation, alignment, and retrieval. Our work is complementary to these directions rather than introducing bidirectional generation itself. In contrast to prior approaches that often focus on tightly coupled bidirectional training or task-specific alignment, we study multimodal biological classification under small, heterogeneous datasets with varying modality availability. The key distinction of our approach lies in its deployment strategy: we treat cross-modal generation as an independently deployable augmentation mechanism, where different generative directions are applied depending on the dataset, and the generated modality is used to support downstream multimodal classification. This perspective emphasizes practical augmentation and modular integration, distinguishing our work from prior alignment- or retrieval-focused methods. Table 1 summarizes capability coverage across models, without implying novelty or priority of invention.

## 3 Method

### 3.1 Overview

Our goal is to improve biological classification by combining generative augmentation and multimodal fusion. Rather than enforcing a single end-to-end mapping between modalities, our approach performs complementary generative augmentations, where either *image-to-text* or *text-to-image*, each enriching one modality using information from the other. The generated samples are incorporated into downstream training as auxiliary inputs, allowing both visual and textual (or tabular) encoders to learn more robust, semantically aligned representations (Fig. 1).

Formally, given paired modalities $x_v$ (image) and $x_t$ (metadata), the framework defines two generators:

$$\tilde{x}_t = \mathcal{G}_{v \to t}(x_v, x_t^{\text{meta}}), \quad \tilde{x}_v = \mathcal{G}_{t \to v}(x_t),$$

where $\mathcal{G}_{v \to t}$ produces textual or structured metadata from visual cues, and $\mathcal{G}_{t \to v}$ synthesizes pseudo-visual samples conditioned on metadata. In the HAM10000 setting, $\mathcal{G}_{v \to t}$ is conditioned on both the dermoscopic

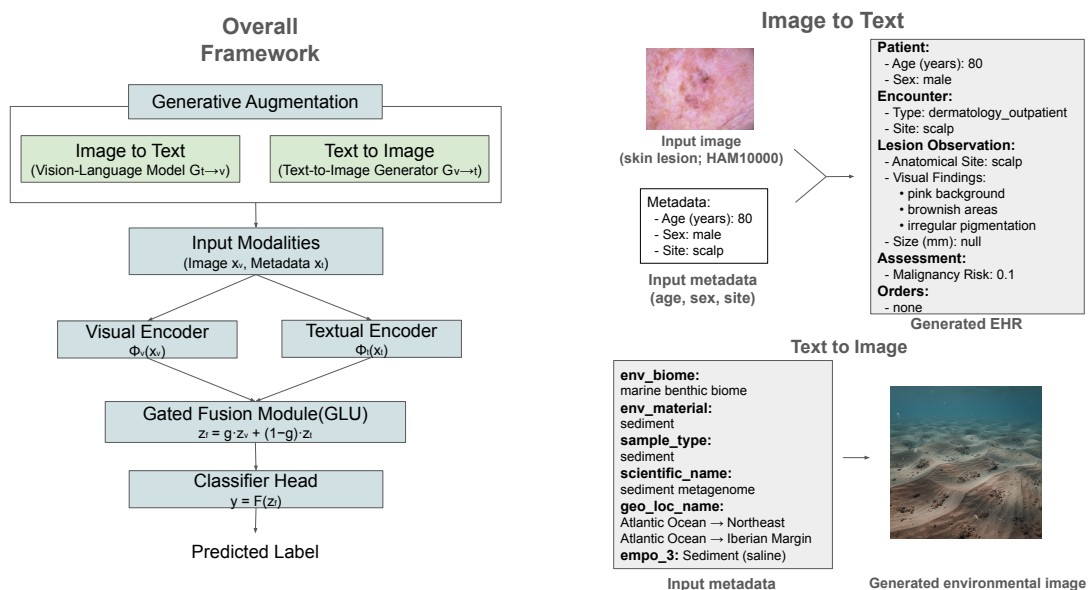

Figure 1: **Overview of the proposed cross-modal generative augmentation framework.** The left panel illustrates the multimodal fusion and classification architecture, which integrates image and metadata inputs through a visual encoder, a textual encoder, and a gated fusion (GLU) module followed by a classifier head. The right panel shows the cross-modal generative augmentation modules: (top) image-to-text generation produces EHR-like diagnostic narratives from dermoscopic images and metadata (HAM10000); (bottom) text-to-image generation synthesizes environmental imagery from tabular metadata (EMPO500). Both generated modalities are incorporated into the shared multimodal classifier, enabling complementary information flow across modalities.

image and available structured metadata (e.g., age, sex, localization). Both generated sets $\tilde{x}_t$ and $\tilde{x}_v$ are used to expand the original dataset. A unified fusion classifier $\mathcal{F}$ then learns from this augmented space through modality-specific encoders $\Phi_v$ and $\Phi_t$, producing the final label prediction $y = \mathcal{F}([\Phi_v(x_v), \Phi_t(x_t)])$. This strategy maintains modularity between the generative and discriminative components while enabling complementary information exchange across modalities. Table 2 summarizes the main symbols used throughout this paper. Unlike tightly coupled or round-trip generative training, which can propagates errors across modalities, our loosely coupled design avoids error amplification while preserving the benefits of cross-modal augmentation. As shown in Appendix E, round-trip generation consistently yielded lower performance than the proposed one-pass augmentation strategy.

## 3.2 Cross-Modal Generative Augmentation

**Image-to-Text (HAM10000).** We use clinical image inputs paired with lesion metadata to generate structured narratives in an EHR-like schema. These textual augmentations, derived from patient and lesion features, mimic diagnostic documentation (e.g., *"pigmented lesion, asymmetric, irregular border"*) and serve as an auxiliary modality for multimodal fusion. Each generated record is vectorized into categorical and numerical embeddings (sex, site, anatomical region, visual findings, etc.), aligned with the real EHR-derived metadata. This module corresponds to the $\mathcal{G}_{v \rightarrow t}$ pathway.

**Text-to-Image (EMPO500).** Conversely, we synthesize images from environmental or biological metadata such as *env_biome*, *env_material*, and *sample_type*. This generation process uses tabular descriptors to condition a diffusion-based generator, producing pseudo-visual textures that capture ecological and environmental cues. The generated samples are used as additional training data for multimodal fusion, expanding the visual feature space and improving generalization under low-resource conditions. This corresponds to $\mathcal{G}_{t \rightarrow v}$ in our formulation.

### 3.3 Multimodal Fusion and Classification

To integrate both visual and tabular (or textual) modalities, we design a lightweight yet expressive **fusion network** composed of three main components: a vision backbone, a tabular encoder, and an adaptive fusion head. The overall framework supports multiple fusion mechanisms; in our experiments, we use concatenation-based fusion for HAM10000 and gated fusion (GLU-style) for EMPO500.

**Visual Encoder.** We adopt a ConvNeXt-v2 (Woo et al., 2023) backbone pretrained on ImageNet-22K/1K (Ridnik et al., 2021) as the visual encoder $\Phi_v$. Given an input image $x_v$, the encoder extracts high-dimensional representations $z_v \in \mathbb{R}^{d_v}$ that capture morphological and structural information. To improve stability under limited biological data, the visual encoder is fine-tuned using dataset-specific training schedules.

**Tabular or Textual Encoder.** For metadata or generated textual representations, we employ a tabular encoder $\Phi_t$ implemented as a multi-layer perceptron (MLP) with learned embeddings for each categorical feature (e.g., sex, anatomical site, or environmental biome). Each categorical feature $x_t^i$ is embedded and concatenated to form the metadata vector $z_t \in \mathbb{R}^{d_t}$:

$$z_t = \text{MLP}\big([\text{Embed}(x_t^1)\|\text{Embed}(x_t^2)\|\cdots\|\text{Embed}(x_t^k)]\big).$$

This representation can flexibly encode either structured EHR metadata (HAM10000) or environmental context (EMPO500).

**Fusion Mechanisms.** For HAM10000, we use a concatenation-based fusion strategy in which visual and tabular embeddings are concatenated and passed through a projection layer before classification. This simpler fusion is sufficient in the dermatology setting, where the visual modality is strong and the structured metadata is relatively well aligned with the image.

For EMPO500, the visual and tabular embeddings are combined via a gated linear unit (GLU)-inspired mechanism (Dauphin et al., 2017; Tsai et al., 2019) that adaptively balances modality contributions:

$$g = \sigma(W_g[z_v\|z_t]), \quad z_f = g \odot z_v + (1 - g) \odot z_t,$$

where $\sigma(\cdot)$ denotes the sigmoid activation and $[\cdot\|\cdot]$ indicates concatenation. This gating allows the model to dynamically emphasize the modality carrying more discriminative cues for each sample. The fused representation $z_f$ is subsequently passed through a two-layer classification head to produce the predicted label:

$$y = \mathcal{F}(z_f) = \text{Softmax}(W_2\,\text{ReLU}(W_1 z_f)).$$

**Evaluation Modes.** The fusion model supports three evaluation modes—*vision-only*, *tabular-only*, and *fusion*—by selectively masking inputs to isolate modality contributions. This enables an interpretable analysis of cross-modal complementarity and quantifies the impact of generated augmentation on performance.

The exact configurations used in all experiments, including fusion type, optimizer, loss functions, and training schedules, are summarized in Section 4.2 (Table 6).

### 3.4 Training Strategy

We employ dataset-specific training configurations for HAM10000 and EMPO500.

For HAM10000, we adopt a two-stage fine-tuning pipeline consisting of multimodal training followed by classifier refinement under class imbalance.

For EMPO500, due to the limited visual signal and the nature of metadata-driven generation, training was performed using a simplified two-stage schedule.

We use dataset-specific augmentation strategies. In HAM10000, Mixup/CutMix is applied only to images and labels during training, while EHR metadata is left unchanged. In EMPO500, we instead use image-only interpolation regularization with probability 0.3; tabular features and class labels are not mixed.

Table 2: Summary of notation.

| Symbol | Description |
|---|---|
| $x_v$ | Input image (visual modality) |
| $x_t$ | Input metadata or text (tabular modality) |
| $x_t^{\text{meta}}$ | Structured metadata used to condition $\mathcal{G}_{v \to t}$ in HAM10000 |
| $\tilde{x}_t$ | Generated text from image-to-text generator $\mathcal{G}_{v \to t}$ |
| $\tilde{x}_v$ | Generated image from text-to-image generator $\mathcal{G}_{t \to v}$ |
| $\Phi_v, \Phi_t$ | Visual and tabular encoders |
| $z_v, z_t, z_f$ | Visual, tabular, and fused embeddings |
| $\mathcal{F}$ | Classification head |
| $g$ | Gating coefficient in GLU fusion |
| $y$ | Predicted label |

Table 3: Performance comparison on the HAM10000 dataset. Results for InceptionV3, ResNet50, DenseNet121, and ALBEF variants are reported from (Adebiyi et al., 2024). Our models (Fusion + EHR-like generation) are implemented and evaluated under the same experimental setting.

| Model | Accuracy | AUC | Macro F1 | Recall | Precision |
|---|---|---|---|---|---|
| *Standard Vision Backbones* | | | | | |
| InceptionV3 | 0.865 | 0.859 | 0.784 | 0.761 | 0.820 |
| ResNet50 | 0.850 | 0.839 | 0.729 | 0.713 | 0.782 |
| DenseNet121 | 0.886 | 0.897 | 0.835 | 0.825 | 0.851 |
| *Multimodal Models (ALBEF)* | | | | | |
| ALBEF Multimodal | 0.941 | 0.914 | 0.903 | 0.852 | 0.896 |
| ALBEF (Adebiyi et al., 2024) Image-only | 0.913 | 0.943 | 0.869 | 0.902 | 0.907 |
| *Our Models (Fusion + EHR-like Report Generation)* | | | | | |
| Fusion (w/o EHR) | 0.885 | 0.980 | 0.818 | 0.783 | 0.864 |
| Meta-only (w/o EHR) | 0.195 | 0.699 | 0.078 | 0.224 | 0.163 |
| Image-only (w/o EHR) | 0.880 | 0.977 | 0.814 | 0.779 | 0.858 |
| Fusion (+EHR-like generation) | **0.959** | **0.993** | **0.910** | 0.892 | **0.937** |
| Meta-only (+EHR-like generation) | 0.295 | 0.933 | 0.353 | 0.481 | 0.504 |
| Image-only (+EHR-like generation) | 0.873 | 0.976 | 0.785 | 0.736 | 0.863 |

Full training details, including optimizer settings, epoch counts, and scheduling configurations, are provided in Appendix C. These configurations reflect the fixed experimental settings used in the reported results, while the framework itself remains modular.

**Calibration and Test-time Augmentation.** Following training, we apply logit bias calibration and temperature sweeping on the validation set to maximize macro-F1, improving robustness to class imbalance. Test-time augmentation (TTA) averages predictions across multiple color-jittered views to further stabilize inference.

**Implementation.** All experiments are conducted on a single NVIDIA A100 GPU using dataset-specific batch size. Training hyperparameters were selected separately for each dataset and are summarized in Table 6. The framework remains fully modular—different encoders or generators can be substituted without retraining the overall system. The code for reproducing all experiments is provided in the supplementary material.

Table 4: Performance comparison on the HAM10000 dataset. Mean and standard deviation are reported as mean(std) over five runs.

| Model | Accuracy | AUC | Macro F1 | Recall | Precision |
|---|---|---|---|---|---|
| Fusion | **0.9570(0.0034)** | **0.9947(0.0011)** | **0.9050(0.0035)** | **0.8900(0.0113)** | **0.9247(0.0145)** |
| Image-only | 0.8714(0.0058) | 0.9739(0.0028) | 0.7755(0.0169) | 0.7474(0.0140) | 0.8388(0.0498) |
| Meta-only | 0.1747(0.1100) | 0.9207(0.0152) | 0.2080(0.1310) | 0.3619(0.1629) | 0.3511(0.1802) |

Table 5: Performance comparison of our models on the EMPO500 dataset. Mean and standard deviation are reported as mean(std) over five runs. Our method employs a text-to-image generation strategy for improved multimodal classification.

| Model | Accuracy | Macro F1 | Precision | Recall | AUC |
|---|---|---|---|---|---|
| *Classical Baselines* | | | | | |
| Logistic Regression (Hastie et al., 2009) | 0.9337(0.0000) | 0.7932(0.0000) | 0.7869(0.0000) | 0.8290(0.0000) | 0.9970(0.0000) |
| Random Forest (Breiman, 2001) | 0.9313(0.0013) | 0.7967(0.0011) | 0.7848(0.0013) | 0.8429(0.0000) | 0.9971(0.0000) |
| SVM (RBF) (Hearst et al., 1998) | 0.9277(0.0000) | 0.7940(0.0000) | 0.7825(0.0000) | 0.8429(0.0000) | 0.9508(0.0081) |
| XGBoost (Chen & Guestrin, 2016) | 0.9223(0.0013) | 0.7346(0.0095) | 0.7189(0.0097) | 0.7845(0.0093) | 0.9965(0.0001) |
| *Our Models (Fusion with Text-to-Image Augmentation)* | | | | | |
| Fusion (Vision+Tabular) | 0.9373(0.0045) | 0.8071(0.0113) | **0.8074(0.0122)** | 0.8370(0.0081) | 0.9698(0.0120) |
| Ensemble (Fusion+MLP) | **0.9386(0.0016)** | **0.8090(0.0059)** | 0.8049(0.0076) | **0.8425(0.0062)** | **0.9974(0.0001)** |
| MLP (Tab-only) | 0.9337(0.0000) | 0.7876(0.0000) | 0.7695(0.0000) | 0.8324(0.0000) | 0.9970(0.0000) |
| Tab-only | 0.9133(0.0045) | 0.7160(0.0176) | 0.6975(0.0178) | 0.7743(0.0201) | 0.9940(0.0061) |
| Vision-only | 0.2319(0.0745) | 0.1753(0.0443) | 0.2300(0.0427) | 0.2289(0.0608) | 0.8987(0.0069) |

## 4 Experiments

We conduct experiments across two complementary biological domains to evaluate whether cross-modal generative augmentation improves multimodal classification performance under data imbalance and limited supervision. All datasets are publicly available and selected to represent distinct modality pairings—image-to-text (HAM10000) and text-to-image (EMPO500). To ensure full reproducibility, all code used in our experiments is included in the supplementary material.

### 4.1 Datasets

#### 4.1.1 Skin Lesion Classification (HAM10000)

The HAM10000 dataset (Adebiyi et al., 2024) consists of 10,015 dermatoscopic images across seven diagnostic categories: *akiec*, *bcc*, *bkl*, *df*, *mel*, *nv*, and *vasc*. The distribution is highly imbalanced (ranging from 115 to 6,705 samples per class), reflecting real-world prevalence in dermatology. Each image is paired with structured metadata, including patient age, sex, and anatomical site. For the image-to-text direction, the generative module $\mathcal{G}_{v \to t}$ produces structured EHR-like captions describing lesion characteristics (e.g., "pigmented lesion with irregular border"). These generated narratives are fused with the visual embeddings using the concatenation-based multimodal classifier described in Section 3. We follow a 70/15/15 train/validation/test split and report results averaged over five random seeds.

#### 4.1.2 Environmental Classification (EMPO500)

The EMPO500 dataset (Shaffer et al., 2022) originates from the Earth Microbiome Project Multi-omics collection (Study ID: 13114). It contains 2,002 microbiome samples annotated with environmental metadata fields such as *env_biome*, *env_material*, and *sample_type*. Each sample is labeled by its *env_feature* category (e.g., *marine benthic feature*, *forest soil*, *saline marsh*), totaling 49 distinct ecological contexts. Due to the small and imbalanced nature of the dataset (3–323 samples per class), we condition a diffusion-based text-to-image generator $\mathcal{G}_{t \to v}$ on tabular metadata to produce pseudo-visual textures representing ecological cues. These generated images expand the training distribution for multimodal fusion, improving robustness

under few-shot settings. Dataset splits follow 1355/315/332 samples for training, validation, and testing, respectively.

Table 6: Summary of experimental configurations used in main results.

|  | HAM10000 | EMPO500 |
|---|---|---|
| Fusion type | Concatenation | Gated (GLU) |
| Backbone | ConvNeXt-v2 (Large) | ConvNeXt-v2 (Base) |
| Optimizer | AdamW | Lion |
| Epochs | 40 + 7 (2-stage) | 30 + 7 |
| Loss | CB-Focal $\rightarrow$ Balanced Softmax | Cross-Entropy |
| Batch size | 24 | 32 |
| Image size | 384 | 224 |
| Mixup | Yes | Yes (p=0.3, image-only) |
| EMA | Yes | No |
| TTA | No | Yes |
| Calibration | Yes (temperature + bias) | No |

## 4.2 Implementation Details

**Experimental setup and evaluation metrics.** All experiments were implemented in PyTorch and executed on a single NVIDIA A100 GPU (80GB VRAM). We fixed all random seeds for reproducibility across NumPy, PyTorch, and CUDA. For multimodal classification, the visual encoder $\Phi_v$ was a ConvNeXt-v2 (Woo et al., 2023) pretrained on ImageNet-22K/1K, and the metadata encoder $\Phi_t$ was a two-layer MLP (512–256 units, ReLU activation, dropout 0.2). Fusion was performed using dataset-specific configurations. For HAM10000, we use concatenation-based fusion followed by a linear projection and softmax classifier. For EMPO500, we use a gated fusion (GLU-style mechanism) to adaptively balance visual and tabular embeddings. We report Accuracy, AUC, macro-F1, Precision, and Recall as main metrics, each averaged over 5 runs with mean $\pm$ standard deviation.

**Training schedule.** We use dataset-specific training configurations. For HAM10000, we adopt a two-stage fine-tuning pipeline with AdamW, consisting of joint multimodal training followed by classifier refinement under class imbalance. For EMPO500, we use a two-stage training schedule consisting of 30 epochs of multimodal training followed by 7 epochs of fine-tuning. Full optimizer, epoch, and scheduling details for each dataset are provided in Appendix C.

For clarity and reproducibility, we summarize the exact configurations used for each dataset in Table 6. While the framework is modular, all results reported in this paper are obtained using these fixed configurations.

**Benchmarking and ablation studies.** For the HAM10000 (image-to-text) task, EHR-like textual reports were generated from dermatoscopic images and used as auxiliary metadata inputs. We compared Image-only, Metadata-only, and Fusion models with and without these generated reports.

For the EMPO500 (text-to-image) task, synthetic pseudo-environmental images were produced from metadata via diffusion-based generation and combined with tabular features using the gated fusion configuration described in Section 3. We compared classical baselines (Logistic Regression, Random Forest, SVM, XGBoost) against our Fusion and Ensemble variants (Table 5).

## 5 Results

### 5.1 Quantitative Results on HAM10000

Table 3 compares our multimodal (Fusion) approach with classical CNN backbones and the pretrained AL-BEF model on the HAM10000 skin lesion benchmark. We distinguish between reported and reproduced

baselines. Results for InceptionV3, ResNet50, DenseNet121, and ALBEF are directly reported from prior work and are included for reference only. All models under "Our Models" are implemented and evaluated within our pipeline using the same data splits, preprocessing, and evaluation metrics to ensure fair comparison. Our Fusion model with EHR-like report generation achieves the highest overall performance with an accuracy of 0.959 and a macro-F1 of 0.910, outperforming both ALBEF Multimodal (0.903) and all vision-only baselines. This demonstrates that integrating textual metadata in an EHR-like generative manner enhances interpretability while improving classification reliability. The improvement over the Image-only variant (0.785 macro-F1) highlights the effectiveness of cross-modal representation learning in capturing diagnostic cues that are not visually evident.

## 5.2 Ablation Study: Effect of Multimodal Fusion

To further assess robustness and stability, we repeated all experiments five times and report mean and standard deviation in Table 4. Across multiple runs, the Fusion model consistently achieves high performance ($0.9570 \pm 0.0034$ accuracy), demonstrating stable convergence and superior generalization compared to the unimodal baselines. While the Meta-only model shows relatively high AUC (0.9207), its classification accuracy remains low (0.1747), indicating that metadata alone are insufficient without image grounding. These results collectively validate that multimodal integration offers complementary gains beyond what either modality can provide independently.

To better understand where the performance gains originate, we analyze improvements across classes grouped by frequency. On HAM10000, we observe that gains are more pronounced in underrepresented classes. For example, rare categories such as "vasc" and "df" show substantially larger improvements in F1 score compared to frequent classes such as "nv". This suggests that cross-modal augmentation helps reduce ambiguity in low-data regimes by providing complementary signals from generated modalities.

Similarly, on EMPO500, improvements are concentrated in several underrepresented environmental categories. Classes with limited samples (e.g., specific sediment or reef-related environments) show notable gains when synthetic visual features are introduced, while frequent classes exhibit smaller relative changes. These results indicate that the primary benefit of cross-modal generation lies in enhancing representation quality under data scarcity rather than uniformly boosting all classes.

Table 7: Comparison between closed-weight and open-weight generators on HAM10000 and EMPO500 datasets.

| Generator Pair | Dataset | Accuracy | Macro F1 | $\Delta$ vs Closed |
|---|---|---|---|---|
| GPT-4o-mini / Gemini-Image | HAM10000 | 0.959 | 0.910 | – |
| Qwen-VL / Qwen-Image | HAM10000 | 0.954 | 0.906 | 0.005 / 0.004 |
| GPT-4o-mini / Gemini-Image | EMPO500 | 0.943 | 0.825 | – |
| Qwen-VL / Qwen-Image | EMPO500 | 0.934 | 0.799 | 0.019 / 0.026 |

## 5.3 Results on EMPO500 Environmental Classification

Table 5 presents the results on the EMPO500 benchmark, which involves multimodal classification across 49 environmental features combining visual and tabular signals. All reported classical baselines are evaluated using the same predefined splits and evaluation metrics within our pipeline to ensure comparability. Our Fusion and Ensemble models achieve competitive performance compared to classical baselines (Random Forest (Breiman, 2001), XGBoost (Chen & Guestrin, 2016), SVM (Hearst et al., 1998), Logistic Regression (Hastie et al., 2009), achieving up to 0.9386 accuracy and 0.8090 macro-F1, while providing modest improvements in class-balanced metrics. The strong performance of Fusion (Vision+Tabular) over Tab-only and Vision-only setups confirms that the cross-modal learning framework effectively captures environmental context and morphological diversity. In particular, the Ensemble model (Fusion+MLP) achieves the best overall results, highlighting the benefit of combining learned multimodal representations with lightweight meta-classifiers. These findings align with our observations on HAM10000, emphasizing that cross-modal

Table 8: Comparison with strong LLM/VLM baselines using prompting strategies. Even with few-shot and Chain-of-Thought prompting, general-purpose models underperform compared to the proposed supervised fusion approach.

| Dataset | Method | Accuracy | Macro-F1 |
|---------|--------|----------|----------|
| HAM10000 | GPT-4o-mini (zero-shot) | 0.1987 | 0.2656 |
|          | GPT-4o-mini (few-shot + CoT) | 0.4516 | 0.3732 |
| EMPO | GPT-4o (zero-shot) | 0.3765 | 0.2846 |
|      | GPT-4o (few-shot + CoT) | 0.6898 | 0.5231 |

integration between visual and structured information is beneficial across biomedical and ecological domains. It is notable that the Fusion and Ensemble modestly improve over MLP tabl-only – this is most likely due to the unique nature of the EMPO500 data which are environmental parameters taken from fresh water systems, an environment where usually little visual information is taken, and therefore, image augmentation has little effect. Thus, it is even more promising that the fusion/ensemble models improve results modestly, showing that even the little information, that exists in pretrained image generators, is helpful.

We additionally compared our approach with several recent large language and vision-language models used as direct classifiers via prompting strategies.

## 5.4 Comparison with LLM/VLM Prompting Baselines

To further contextualize our results, we compare the proposed model against strong LLM/VLM baselines used as direct classifiers, including GPT-4o(-mini) with zero-shot, few-shot, and Chain-of-Thought (CoT) prompting, as well as BiomedCLIP. As shown in Table 8, general-purpose prompting-based approaches perform substantially worse than the proposed supervised multimodal fusion model across both datasets. While few-shot and CoT prompting significantly improve LLM performance, a clear gap remains compared to task-specific supervised training. This gap is particularly pronounced on the EMPO dataset, which involves highly imbalanced classes and structured metadata. These results highlight that, despite recent advances in LLM/VLM reasoning, direct prompting is insufficient for complex multimodal biological classification, and dedicated multimodal fusion remains essential. Full comparisons, including additional prompting variants, are provided in Appendix H.

## 5.5 Complementary Ensemble Evaluation

Although the ensemble approach is not part of our core framework, we include it as a complementary evaluation to assess performance stability in multimodal classification. By averaging the predictions from the Fusion classifier and a standalone MLP, we observed consistent yet modest improvements across all metrics (Table 5). This simple ensemble reached 0.9386 accuracy and 0.8090 macro-F1 on EMPO500, suggesting that ensembling can further stabilize multimodal prediction, particularly under limited-sample conditions.

## 5.6 Model-Agnostic Verification

To validate the model-agnostic nature of our cross-modal generative augmentation framework, we replaced the closed-weight generators used in the main experiments (GPT-4o-mini (Hurst et al., 2024) for image-to-text and Gemini 2.5 Image (Comanici et al., 2025) for text-to-image) with open-weight counterparts, Qwen-VL (Bai et al., 2023; Wang et al., 2024; Team, 2025), and Qwen-Image (Wu et al., 2025). Both replacements were applied without modifying the fusion or classifier modules. As shown in Table 7, the open-weight configuration achieved comparable results on both datasets, with only marginal performance differences ($\Delta$ Accuracy $< 0.5\%$ and $\Delta$ F1 $< 0.4\%$ in HAM10000 dataset). These results confirm that our framework is not tied to any specific proprietary model and remains reproducible using publicly available generators.

The relatively small performance gap between closed- and open-weight generators suggests that strict perceptual fidelity is not the dominant factor driving downstream improvements. Instead, the observed gains appear to correlate more strongly with semantic alignment between the generated modality and its conditioning inputs. This interpretation is consistent with the generation-quality analysis in Appendix K, where samples with higher alignment scores yield larger multimodal performance gains. The robustness across different generator families further suggests that semantically consistent complementary signals are more important than generator-specific visual fidelity.

## 5.7 Generative Output Consistency Evaluation

To assess the reliability of the generated modalities used in our framework, we conduct an automatic consistency evaluation of both EHR-like narratives produced from dermoscopic images (image-to-text) and environmental images synthesized from tabular metadata (text-to-image). Since human evaluation at scale is costly and domain-dependent, we adopt the increasingly common paradigm of LLM-as-a-judge, supported by recent work showing that large language models can serve as stable, high-agreement evaluators across multimodal and NLG tasks (Zheng et al., 2023; Kocmi & Federmann, 2023; Liu et al., 2023). We use GPT-4o as the multimodal evaluator, following protocols similar to G-Eval (Liu et al., 2023). The full evaluation prompts and scoring instructions are provided in Appendix B for reproducibility. In addition, expert validation was conducted to assess the clinical plausibility of generated EHR narratives, with the evaluation protocol and results provided in Appendix I.

**EHR Consistency (Image-to-text).** We randomly sample 120 generated EHR JSON records from HAM10000, evenly distributed across diagnostic categories. GPT-4o assesses the medical alignment between each generated narrative and its ground-truth diagnosis using a 1–5 scale, where higher scores indicate stronger diagnostic consistency. The generated narratives achieve an average alignment score of 4.40/5, demonstrating that they capture key clinical attributes such as pigmentation patterns, border irregularities, and malignancy-related morphological cues.

**Environmental Image Consistency (Text-to-image).** For EMPO500, we evaluate 266 generated environmental images across 50 ecological classes. GPT-4o measures visual alignment between each synthesized image and its conditioning metadata (e.g., biome, salinity, sample type, environmental material). The synthetic images obtain an average alignment score of 4.56/5, indicating strong metadata-to-visual grounding despite the inherent difficulty of ecological texture generation.

To assess potential evaluator bias, we additionally compare GPT-4o with an independent evaluator (Gemini-2.5-Flash) on 120 samples. The two evaluators show strong agreement (Pearson 0.68, Spearman 0.76, 76% exact match), suggesting that the evaluation is not substantially driven by evaluator-specific bias (see Appendix L).

To further examine whether generation quality translates to downstream performance, we observe that higher alignment scores are associated with more reliable fusion gains, while this relationship is weak for image-only models, indicating that generation quality directly affects multimodal utility (see Appendix K).

## 5.8 Qualitative Interpretability Analysis

To further examine the interpretability of our cross-modal generative augmentation framework, we present representative success cases where multimodal fusion enables clinically or scientifically meaningful interpretation beyond unimodal approaches. Figure 2 summarizes two qualitative examples demonstrating how interpretability arises directly through the generated modalities themselves.

In Figure 2(a), the fusion model correctly identifies the lesion as *benign keratosis (bkl)*, whereas the vision-only and metadata-only models fail—one overestimating malignancy (*akiec*) and the other confusing it with a benign *nevus (nv)*. The generated EHR provides explicit morphological evidence, describing *brownish pigmentation*, *irregular borders*, and a *network pattern*, while assigning a low malignancy risk (0.1), consistent with the clinical appearance of seborrheic keratosis. Here, interpretability emerges from the textual generation itself, which exposes a human-readable reasoning trace aligned with clinical knowledge.

**(a)** **(b)**

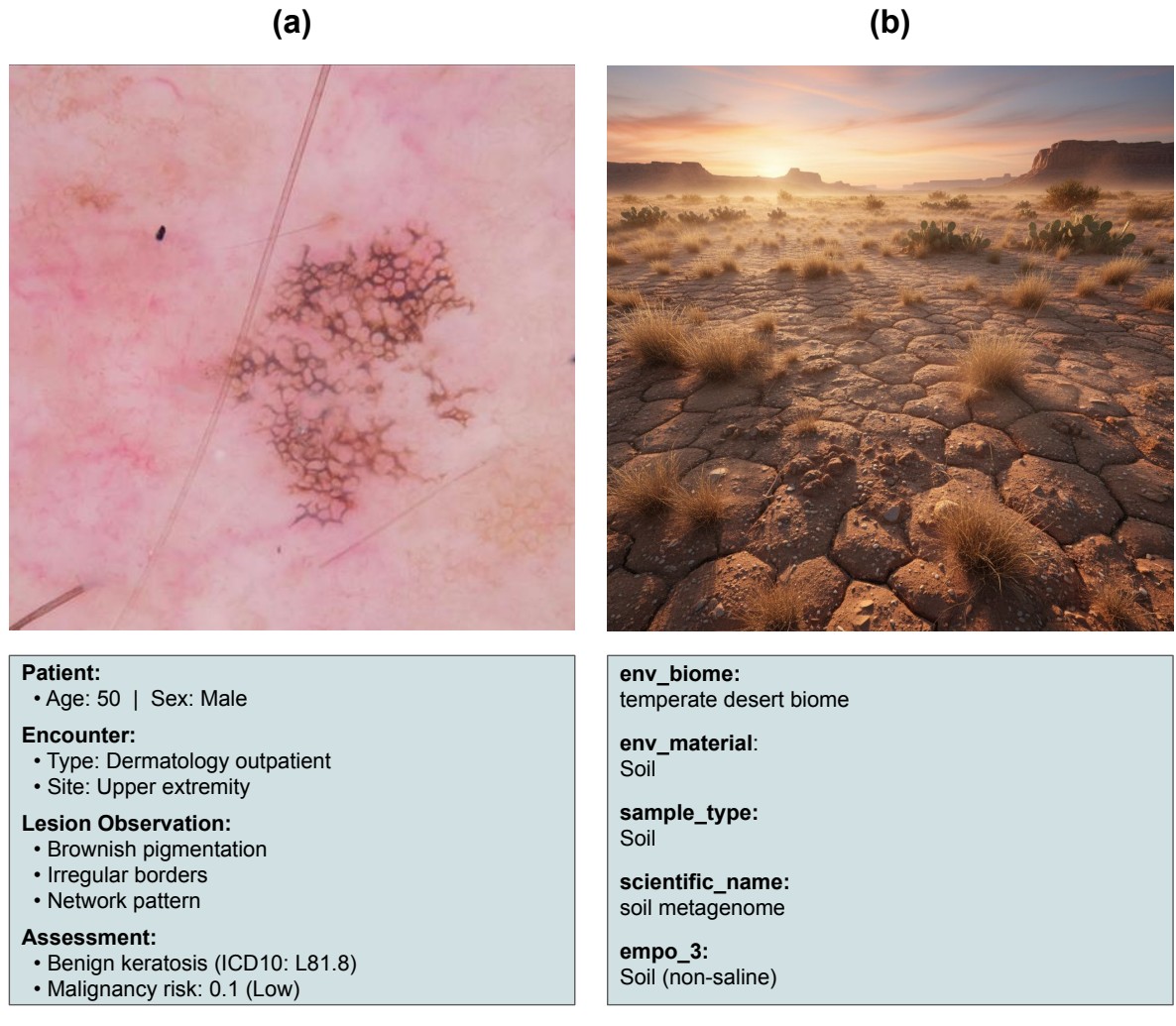

Figure 2: **Representative fusion success cases illustrating interpretability.** (a) *HAM10000 (Image→EHR):* The fusion model's correct prediction of *benign keratosis* aligns with generated structured findings (*brownish pigmentation*, *irregular borders*, low-risk assessment), where the textual EHR output itself provides a transparent clinical rationale. (b) *EMPO500 (Metadata→Image):* The fusion model integrates environmental metadata (*temperate desert biome*, *soil sample*) to correctly infer *desert sand*, while unimodal baselines misclassify it as marine or anthropogenic. Generated image visually grounds the reasoning process by reflecting terrestrial texture patterns.

In Figure 2(b), the fusion model integrates structured environmental metadata (*temperate desert biome*, *soil sample*) to correctly infer the label *desert sand*, while uni-modal baselines misclassify the same sample as a *marine benthic feature* (image-only) or an *anthropogenic contamination feature* (tabular-only). The generated image captures sandy terrestrial textures distinct from marine patterns, providing a visual rationale for the model's prediction. In this case, interpretability emerges from the visual generation, which grounds environmental context through image formation. These examples suggest that cross-modal generation can enhance interpretability not only through model alignment, but also through the generated modalities themselves—textual in clinical data and visual in environmental data.

However, we emphasize that the generated modalities should not be interpreted as faithful or causal explanations of model predictions. While they can provide plausible and human-readable signals aligned with the prediction, they may also reflect spurious correlations or generator biases.

# 6 Conclusion

Across both datasets, the proposed framework demonstrates consistent improvements through cross-modal integration, where dataset-specific generation is applied depending on modality availability. In the clinical domain (HAM10000), the model benefits from incorporating metadata and structured descriptors to enrich lesion understanding. In contrast, in the ecological EMPO500 dataset, generating synthetic visual representations from metadata improves diversity and helps the model generalize across heterogeneous habitats. Together, these experiments indicate that multimodal learning pipelines can bridge domain-specific data gaps by synthesizing complementary modalities, yielding improved predictive accuracy and enhanced human-readability of auxiliary multimodal signals. Future work will explore scaling this cross-modal generation paradigm to larger biomedical datasets and investigating its potential for real-world deployment in clinical and environmental monitoring systems.

Project code is publicly available at: `https://github.com/EESI/cross-modal-biological-augmentation`

# 7 Broader Impact

This work explores cross-modal generative augmentation for multimodal biological classification. While the proposed framework improves performance under limited data settings, it also introduces potential risks that should be considered. First, the use of pre-trained generative models may propagate biases present in the underlying training data. Such biases could affect both generated modalities and downstream classifiers, particularly in sensitive biological or clinical applications. Second, generated modalities may contain inaccuracies or hallucinated content. When such synthetic information is incorporated into the classification pipeline, it may introduce misleading signals and degrade reliability if not properly controlled. Finally, although our framework is designed for research purposes, its direct deployment in real-world biological or clinical decision-making systems requires careful validation. Future work should explore robust filtering, uncertainty estimation, and domain-specific validation to mitigate these risks. In medical settings, generated EHR-like text may appear clinically plausible even when the underlying prediction is incorrect, potentially inducing over-trust from users. Similarly, in environmental applications, generated images may look visually convincing while remaining only weakly faithful to the underlying metadata. Therefore, generated cross-modal outputs should not be treated as verified explanations, especially in high-stakes settings.

## Acknowledgement

This work has beensupported by both 1) the U.S. National Science Foundation grant #2107108 and 2) the E.U. project MIS 5154714 of the National Recovery and Resilience Plan Greece 2.0 funded by the NextGenerationEU Program.

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

# A   Datasets

We use publicly available and lightweight datasets to ensure reproducibility and accessibility.

## A.1   HAM10000 Dataset

A dataset of 10,015 dermatoscopic images across seven diagnostic categories, with class imbalance reflecting real-world prevalence. The distribution of samples across train, validation, and test splits is summarized in Table 9.

## A.2   EMPO500 Dataset

A collection of microbiome environmental metadata samples, labeled with ecological context (e.g., host-associated, saline water, soil). We used Earth Microbiome Project Multi-omics (EMP500) dataset (Study ID: 13114) (Shaffer et al., 2022). Table 10 provides a detailed breakdown of sample counts across environmental features and dataset splits.

| Class | Train | Val | Test | Total |
|-------|-------|-----|------|-------|
| akiec | 235 | 41 | 51 | 327 |
| bcc | 355 | 82 | 77 | 514 |
| bkl | 734 | 180 | 185 | 1099 |
| df | 75 | 17 | 23 | 115 |
| mel | 775 | 193 | 145 | 1113 |
| nv | 4728 | 958 | 1019 | 6705 |
| vasc | 85 | 41 | 16 | 142 |
| **Total** | **6987** | **1512** | **1516** | **10015** |

Table 9: HAM10000 Dataset statistics: number of samples per class across Train/Validation/Test splits.

---

**EHR Generation Prompt (Qwen2.5-VL)**

**System Prompt**
You are a clinical documentation assistant for dermatology research. Extract structured EHR-like JSON data from the dermoscopic image and metadata. Output **only valid JSON** strictly following the schema below. Avoid repeating the same phrases or findings; each field should be concise and unique. If uncertain, use "unknown".

**JSON Schema**

```
{
  "patient": {
    "age_years": "integer",
    "sex": ["male", "female", "unknown"]
  },
  "encounter": {
    "encounter_type": "string",
    "site": "string"
  },
  "lesion_observation": {
    "anatomical_site": "string",
    "visual_findings": ["string"],
    "size_mm": "number",
    "image_quality_note": "string"
  },
  "assessment": {
    "provisional_diagnosis_label": "string",
    "malignancy_risk": "number",
    "rationale": ["string"]
  }
}
```

**User Prompt Template**

```
Metadata:
- age: <AGE>
- sex: <SEX>
- localization: <LOCALIZATION>
- dataset_label_dx (for reference only): <DX>

Please extract structured JSON based on the image and metadata.
```

Figure 3: Prompt used for generating structured EHR-like records with Qwen2.5-VL.

## B  Prompts

### B.1  EHR Generation Prompt

We provide below the exact system and user prompts (Fig. 3) used to generate structured EHR-like JSON records from dermoscopic images using VLMs such as Qwen2.5-VL and gpt-4o-mini.

### B.2  Environmental Image Generation Prompt

The following prompt (Fig. 4) was used to synthesize photo-realistic environmental scenes conditioned on EMPO3 and associated metadata with Qwen-Image and Gemini 2.5 Flash Image.

---

**Environmental Image Generation Prompt**

**Prompt Template**

```
A highly detailed, photo-realistic scene of a natural environment
representing the biome '<env_biome>', showing the material '<env_material>'
and sample type '<sample_type>' in its natural context. The setting reflects
the scientific sample '<scientific_name>' categorized under EMPO3 '<empo_3>'.

Focus on environmental textures, terrain, and lighting -
no human-made objects, no laboratory tools, and no text or labels visible.
```

Figure 4: Full prompt used for environment image generation.

---

**EHR–Diagnosis Alignment Evaluation Prompt**

```
You are a clinical evaluation assistant.
Evaluate how well the generated EHR JSON aligns with the TRUE dermatology
diagnosis label.

Your job is to judge ONLY the medical alignment of the EHR with the true
diagnosis. Do NOT evaluate image quality, language quality, schema correctness,
or style.

Rate alignment on a 1-5 scale:
1 = EHR is medically inconsistent or contradicts the true diagnosis
2 = Mostly inconsistent
3 = Partially aligned (some correct signals but overall weak match)
4 = Mostly aligned (diagnosis and clinical description largely consistent)
5 = Strongly aligned (EHR strongly supports or accurately reflects the
    true diagnosis)

Return ONLY a JSON:
{"score": <1-5 integer>, "reason": "<short explanation>"}
```

Figure 5: Prompt used for evaluating medical alignment between generated EHRs and true dermatology diagnoses.

### B.3 EHR–Diagnosis Alignment Evaluation Prompt

To assess how well each generated EHR record reflects the true dermatological diagnosis, we use the following GPT-4o evaluation prompt (Fig. 5). The model is instructed to judge only medical alignment, ignoring style, schema quality, or formatting.

### B.4 EMPO500 Visual Alignment Evaluation Prompt

To automatically assess the logical consistency between a generated image and its conditioning metadata, we use the following multimodal evaluation prompt (Fig. 6) with GPT-4o.

## C   Metadata Encoding and Training Details

### C.1   Metadata Encoding

Each EHR-like record is converted into a fixed-length metadata vector composed of (1) numerical variables, (2) one-hot categorical variables, and (3) multi-hot visual-finding features. Numerical fields include patient

---

**Visual Alignment Evaluation Prompt**

```
You are an expert evaluator for AI-generated images. You will be given a text
prompt (metadata) and an image that was generated based on that text.

Your task is to evaluate the logical consistency and alignment between the text
and the image. Does the image visually represent the key information described
in the text?

Text Metadata:
{dialogue}

---
Instructions:
Rate the alignment on a scale from 1 to 5.
1: Poor. The image completely ignores or contradicts the text metadata.
3: Moderate. The image reflects some elements of the metadata but misses key
   aspects or includes incorrect elements.
5: Excellent. The image is a perfect and logical visual representation of all
   key elements in the text metadata.

Provide your score as a single number (e.g., 4.5)

Score:
```

Figure 6: Prompt used for assessing image–metadata logical consistency using GPT-4o.

age, lesion size (mm), and the malignancy risk score. Age and lesion size are standardized using the training-set mean and standard deviation, while the malignancy risk score is used as a raw scalar feature. Categorical attributes—patient sex, encounter type, encounter site, and anatomical site—are encoded using one-hot vectors, with an explicit `unknown` slot for unseen values. The "orders" field (e.g., biopsy, urgent referral, reassurance) is mapped to a fixed 5-dimensional one-hot vector. Visual findings are tokenized using simple lowercase string matching, and the 64 most frequent tokens in the training split are used to form a multi-hot representation. All components are concatenated to form the final metadata vector used by the tabular encoder.

## C.2 Training Procedure

We provide dataset-specific training configurations for HAM10000 and EMPO500. All experiments are conducted on a single NVIDIA A100 GPU with deterministic seeds.

**HAM10000 Training.** In **Stage 1**, we jointly fine-tune the visual backbone, metadata encoder, and fusion head for 40 epochs using AdamW (learning rates $3 \times 10^{-5}$ for backbone parameters and $3 \times 10^{-4}$ for fusion/meta parameters). A 4-epoch warmup is followed by cosine learning-rate decay. Class-balanced focal loss ($\gamma = 1.5$) is used, with effective class weights computed from the empirical label distribution, and mixup/CutMix augmentation is applied during training. Exponential moving average (EMA) is maintained with decay 0.9999.

In **Stage 2**, the backbone is kept frozen and only the metadata encoder and classifier head are optimized for 7 additional epochs using Balanced Softmax cross-entropy with AdamW (learning rate $2 \times 10^{-4}$). This stage primarily refines behavior under label imbalance by reweighting logits according to class frequencies.

For augmentation, we apply batch-level Mixup/CutMix only to the image branch using the `timm` implementation. The EHR-derived metadata vectors are not mixed; instead, the mixed image is paired with the original metadata vector while supervision is provided using soft labels.

**EMPO500 Training.** For EMPO500, we adopt a simplified training scheme due to the metadata-driven nature of the task. The multimodal model is trained end-to-end using Lion with cosine learning rate scheduling, without staged freezing. Other EMPO500 training settings follow the dataset-specific configuration summarized in Table 6 unless otherwise specified.

Unlike HAM10000, we do not use label-level Mixup for EMPO500. Instead, with probability 0.3, we apply image-space interpolation regularization by linearly mixing each image with another image in the same mini-batch, while leaving the tabular input and class label unchanged. This acts as a lightweight visual regularizer rather than full multimodal Mixup.

### C.3 Calibration and Inference

For the HAM10000/EHR pipeline, after training we apply post-hoc calibration on the validation set. Temperature scaling is tuned by sweeping $\tau$ over $\{0.0, 0.25, \ldots, 2.0\}$ and selecting the value that yields the highest macro-F1. A per-class bias vector is then optimized via coordinate descent using candidate adjustments in $\{-1.5, -1.0, -0.5, 0, 0.5, 1.0, 1.5\}$. The final model uses EMA weights together with the optimized temperature and bias adjustments during inference. No test-time augmentation is applied to HAM10000 unless otherwise specified.

## D  Qualitative Failure Cases

To complement the success cases in Section 2, we present two representative failure cases where cross-modal generative augmentation produces plausible outputs that are nevertheless inconsistent with the ground-truth labels.

In Figure 7(a), the ground-truth diagnosis is *benign keratosis (bkl)*, but the fusion, image-only, and metadata-only classifiers all predict *nevus (nv)*. The generated EHR describes a 50-year-old male with a "brownish lesion" with "irregular borders" on the upper extremity, assigns a low malignancy risk score (0.1), and lists seborrheic keratosis as the top structured diagnosis candidate. Thus, the text-only reasoning trace appears clinically coherent and label-consistent, yet the final predicted class is wrong. This mismatch illustrates that fluent, structured EHR generation does not guarantee that the underlying decision boundary is correct, and highlights the need for calibrated outputs and expert oversight.

In Figure 7(b), the EMPO500 metadata describe a *subpolar coniferous forest biome* with a lichen soil sample and an *env_feature* of *bog*, but the fusion and tab-only models both predict *cliff*, while the image-only model misclassifies the scene as a *marine benthic feature*. The generated image resembles a mossy forest floor with shallow water and tree trunks, which is broadly compatible with a bog-like environment but contains ambiguous textures that the visual backbone appears to over-associate with marine substrates. This case shows that visual interpretability via generation can still be driven by biased priors rather than faithfully reflecting the underlying environmental conditions.

These failure cases emphasize that cross-modal generative augmentation should be viewed as a tool for revealing model behavior rather than as a guarantee of correctness. The generated text and images can may produce convincing but incorrect outputs because they are fluent and visually realistic.

## E  Additional Ablation: Round-trip Cross-Modal Generation

Our main experiments focus on "one-pass" cross-modal generation, where we either (i) generate EHR-like text from real images and fuse it with the original image (HAM10000), or (ii) generate images from structured metadata and fuse them with tabular features (EMPO500). A natural question is whether chaining generators in a "round-trip" fashion—image→text→image or text→image→text—can further improve robustness or calibration, or whether such chaining instead amplifies errors across modalities.

**HAM10000: image→text→image.** For HAM10000, we start from real dermoscopic images, generate structured EHR-like records (image→text), and then synthesize new dermoscopy-style images conditioned

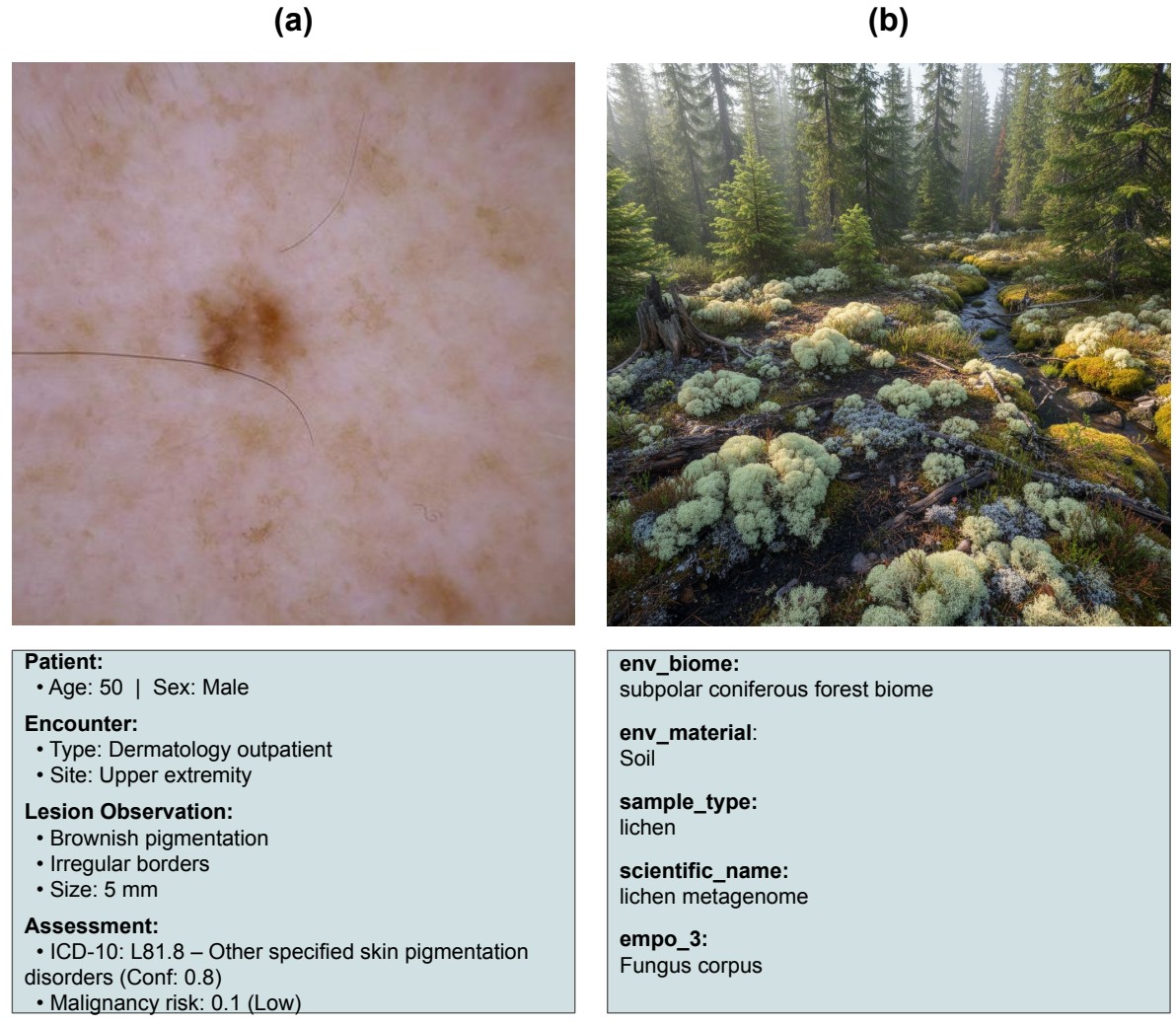

Figure 7: **Representative failure cases illustrating the limits of cross-modal generative augmentation.** (a) *HAM10000 (Image→EHR):* all three classifiers (fusion, image-only, metadata-only) predict an incorrect label, even though the generated EHR narrative is clinically consistent with benign keratosis. (b) *EMPO500 (Metadata→Image):* the fusion and tab-only models predict an environmentally inconsistent label (*cliff*) and the image-only model predicts a *marine benthic feature*, despite metadata describing a bog-like forest environment.

on these EHRs (text→image). We augment the training set with the EHR-conditioned synthetic images and retrain the same fusion + meta-MLP ensemble as in the main experiment. On the test set, the round-trip model achieves accuracy 0.946, macro-F1 0.885, and AUROC 0.984. While these numbers remain strong, they are slightly *worse* than our best one-pass image→text fusion baseline (Table 4). Qualitatively, many EHR-conditioned synthetic images lack fine-grained dermoscopic structure and realistic color/texture variation, behaving more like noisy views than clinically faithful lesions; as a result, they do not provide additional signal beyond the original images and sometimes inject label noise.

**EMPO500: text→image→text.** For EMPO500, we symmetrically explore a text→image→text loop. We first generate synthetic habitat images from ground-truth EMPO-style metadata, then apply an LLM-based captioner to recover metadata-like descriptions from these images. We then train (i) an image+metadata fusion model and (ii) a tab-only MLP using a concatenation of ground-truth and LLM-derived

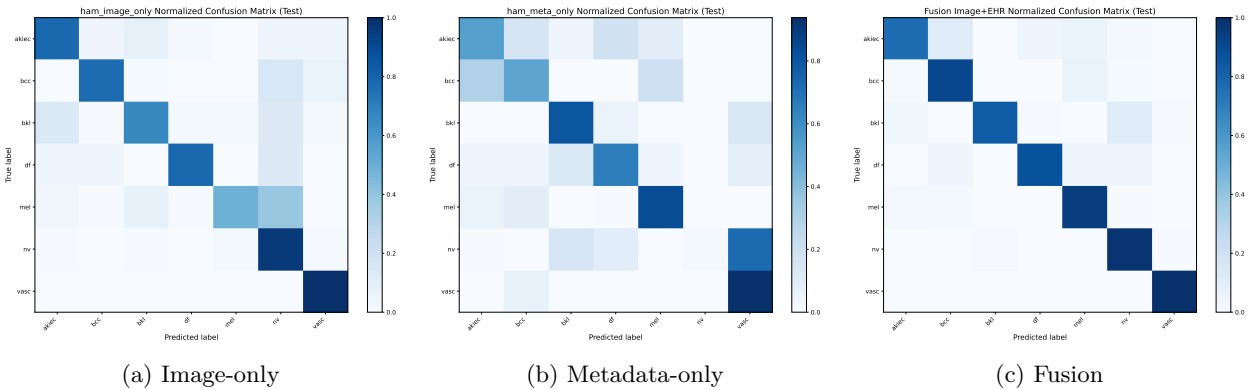

(a) Image-only        (b) Metadata-only        (c) Fusion

Figure 8: Class-wise confusion matrices on HAM10000 under image-only, metadata-only, and fusion settings. The fusion model shows improved concentration around the diagonal, indicating stronger class discrimination.

metadata features, along with their ensemble. The best fusion model reaches accuracy 0.925, macro-F1 0.757, macro-precision 0.737, macro-recall 0.806, and UROC 0.996; the tab-only MLP and the fusion+MLP ensemble reach accuracy 0.931 with macro-F1 $\approx$ 0.79–0.78 and AUROC 0.997. These round-trip scores are noticeably lower than our one-pass text→image fusion benchmark (Table 5), especially on rare habitat classes.

Qualitatively, EMPO500 exposes an important limitation of round-trip generation: the label and feature spaces are extremely fine-grained and ontology-sensitive. Image-conditioned metadata often uses plausible but non-canonical habitat names, merges or splits categories, or drifts away from the EMPO ontology (Shaffer et al., 2022). This semantic drift breaks the alignment between generated metadata and the target label space, so that adding LLM-derived EMPO features behaves more like structured label noise than useful augmentation.

These ablations suggest that fully round-trip cross-modal generation (image→text→image or text→image→text) is *not* universally beneficial in our settings. Chaining generators across modalities can introduce compounding errors and ontology drift, particularly in high-cardinality ecological label spaces. For this reason, we treat round-trip variants as exploratory negative results and retain the one-pass bidirectional models as the primary components of our benchmark.

# F  Class-wise Confusion Matrix Analysis

To further examine whether multimodal fusion improves class discrimination, we compare class-wise confusion matrices for the HAM10000 and EMPO datasets under three settings: image-only, metadata-only (or tab-only), and fusion. Overall, the fusion models show a clearer concentration on the diagonal, indicating better alignment between true and predicted labels and more balanced use of complementary information across modalities.

## F.1  HAM10000

Figure 8 compares the confusion matrices on HAM10000. The image-only model captures several visually distinctive classes, but still shows noticeable confusion among clinically similar categories. The metadata-only model provides useful signals for some classes, yet its predictions are less consistently aligned with the ground-truth labels. In contrast, the fusion model yields a cleaner class-wise separation overall, suggesting that combining image features with EHR-style metadata improves robustness and reduces ambiguity across lesion categories.

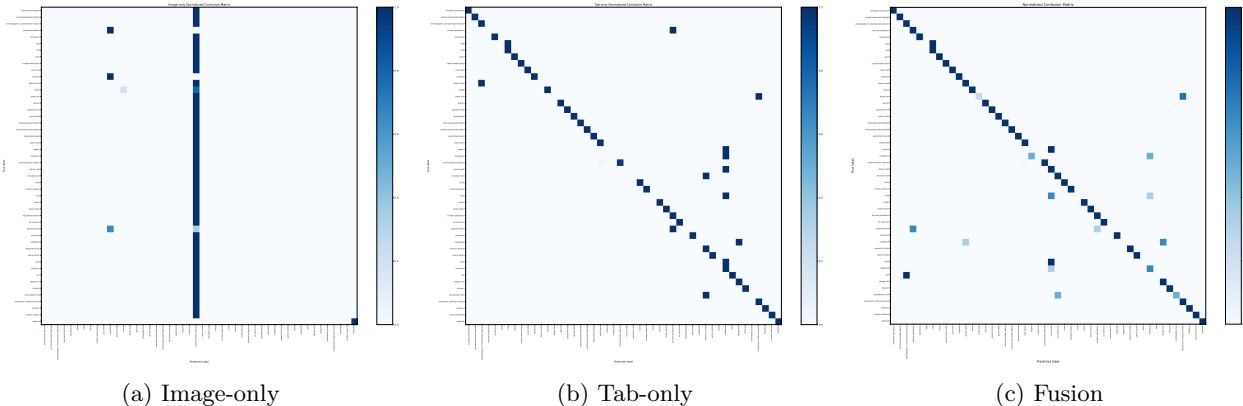

| (a) Image-only | (b) Tab-only | (c) Fusion |

Figure 9: Class-wise confusion matrices on EMPO under image-only, tab-only, and fusion settings. Fusion produces a more consistent diagonal structure than the unimodal baselines, indicating complementary gains from multimodal integration.

## F.2 EMPO

Figure 9 shows the same comparison on EMPO, which is more challenging due to its broader label space and visually less distinctive categories. The image-only model collapses many samples into a limited subset of predictions, indicating weak separability from visual information alone. The tab-only model performs substantially better and already forms a strong diagonal structure, reflecting the importance of structured metadata for this task. The fusion model further sharpens this pattern and reduces several off-diagonal errors, showing that image cues still provide additional value when combined with tabular context.

These qualitative results support the quantitative findings: fusion does not simply average two modalities, but improves class-wise decision boundaries by leveraging complementary information from both image and structured metadata.

## G Comparison with a Simple Concatenation Baseline

To further evaluate the effectiveness of the proposed fusion design, we compare it against a simple concatenation baseline in which image and metadata representations are directly concatenated before classification. This comparison is intended to isolate whether the gains come merely from combining modalities, or from the proposed fusion mechanism itself. We report the final test-set results on both HAM10000 and EMPO.

As shown in Table 11, the proposed method generally achieves higher accuracy and macro-F1 than the simple concatenation baseline on both datasets, indicating that its benefits are not solely due to multimodal input combination. On HAM10000, the proposed method also achieves higher AUROC, suggesting better overall ranking quality in addition to stronger class-balanced classification. On EMPO, the proposed method substantially improves accuracy, macro-F1, macro-precision, and macro-recall, although the concatenation baseline attains a higher AUROC. These observations suggest that the relative advantages of each fusion strategy may vary depending on the dataset characteristics and evaluation metric. Since these results are based on a single experimental run, we report both approaches to provide a more comprehensive comparison of multimodal fusion behaviors across datasets.

## H Comparison with LLM/VLM Baselines

To further contextualize the performance of the proposed multimodal fusion model, we compare it with several recent large language and vision-language models used as direct classifiers. Specifically, we evaluate Qwen2.5-VL, GPT-4o(-mini), BiomedCLIP, and BioMistral under direct prompting or zero-shot inference settings.

These models are used without task-specific training and therefore represent strong general-purpose baselines. For completeness, we also report the performance of our supervised multimodal fusion model on the same test sets.

Table 12 summarizes the results. Overall, direct prompting or zero-shot VLM approaches perform substantially worse than the proposed supervised fusion method. The gap is particularly pronounced for the EMPO dataset, which involves a large number of highly imbalanced classes. These results highlight the importance of task-specific multimodal fusion and supervised training for structured biomedical and environmental classification tasks.

## I  Clinical Plausibility Validation of Generated EHR

While the main evaluation focuses on internal consistency and modality alignment, we additionally conducted a small-scale expert validation to assess the clinical plausibility of the generated EHR narratives.

Two licensed physicians (MDs) independently reviewed a total of 70 randomly sampled generated EHR cases drawn across multiple diagnostic categories. For each case, the reviewers assessed two criteria: (1) whether the clinical description (e.g., morphology, clinical history, and physical examination) logically matches the provided diagnosis, and (2) whether the note is internally consistent and free of obvious structural contradictions (e.g., mismatches between demographic attributes and narrative content). Each criterion was evaluated using a binary judgment (Yes/No).

Table 13 summarizes the aggregated results of the expert review. Overall, the large majority of generated EHR narratives were judged to be clinically plausible and internally coherent. Specifically, 67 out of 70 cases (95.7%) were judged to have diagnosis-consistent clinical descriptions, and 64 out of 70 cases (91.4%) were considered logically consistent without structural contradictions.

Although this validation is limited in scale and does not constitute a formal clinical study, the results provide preliminary evidence that the generated EHR narratives are not only linguistically coherent but also medically reasonable.

## J  Additional Experiments on PAD-UFES-20

We provide additional experiments on the PAD-UFES-20 dataset to further evaluate the effectiveness of our framework under a unified protocol (patient-level split, 5 runs with mean $\pm$ std). We investigate both generative directions: (i) image-to-text and (ii) text-to-image.

### J.1  Image-to-Text Direction (Image → Generated Metadata)

This experiment demonstrates how generating synthetic clinical metadata from images enriches the multimodal representation. Our proposed method shows consistent improvements across all evaluation metrics.

### J.2  Text-to-Image Direction (Metadata → Synthetic Image)

This experiment evaluates how synthesizing images from clinical metadata provides visual augmentation. In this protocol, generated images are used as the visual modality conditioned on metadata, without using the corresponding real dermoscopic images during multimodal training. As shown in Table 15, while the Random Forest baseline achieves high overall accuracy, it exhibits relatively limited performance on class-balanced metrics in this imbalanced biomedical setting.

Although accuracy decreases, the improvements in Macro-F1 and Macro-Recall indicate that our method better captures minority classes, which is crucial in imbalanced biomedical settings.

We intentionally do not adopt a tightly coupled or round-trip training strategy. As shown in Appendix E, round-trip generation leads to degraded performance due to error accumulation, which motivates our design choice of using independent augmentation pathways.

### J.3 Gap Between Synthetic and Real Multimodal Supervision

To better quantify how closely generated modalities approximate true multimodal supervision, we compare three settings on PAD-UFES-20 under the same multimodal architecture: (i) metadata-only input, (ii) metadata combined with generated images synthesized from metadata, and (iii) metadata combined with real dermoscopic images.

All experiments use the same fusion model and training configuration to ensure fair comparison.

Synthetic image augmentation substantially improves over metadata-only learning, but still underperforms true multimodal supervision with real paired images (Table 16). These results suggest that generated modalities provide meaningful complementary signals, while also highlighting the remaining gap between synthetic modality reconstruction and real multimodal observations.

### J.4 Comparison with Conventional Image Augmentation

To evaluate whether the observed gains arise simply from increased augmentation, we compare metadata-conditioned synthetic image augmentation against conventional image augmentation strategies on PAD-UFES-20.

We compare three settings: (i) the original multimodal setting, (ii) standard image augmentation using RandomResizedCrop, RandAugment, and RandomErasing, and (iii) synthetic image augmentation generated from metadata using the proposed framework.

In the synthetic augmentation setting, generated images are introduced as additional training samples conditioned on metadata, rather than reconstructed replacements of the original dermoscopic images, and no feature-level fusion is performed between real and generated images of the same instance.

The standard augmentation baseline primarily applies distribution-preserving perturbations to existing images, whereas the proposed approach introduces metadata-conditioned semantic expansion through cross-modal generation (Table 17). The stronger performance of synthetic augmentation suggests that the gains are not solely attributable to increased data quantity or conventional augmentation effects, but rather to semantically grounded complementary information introduced by the generated modality.

## K Detailed Analysis of Generation Quality and Downstream Impact

To further investigate the relationship between generation quality and downstream performance, we conduct a sample-level analysis on the HAM10000 dataset. Each generated EHR is assigned an alignment score (1–5) based on GPT-4o evaluation, and this score is linked to the corresponding classification outcome.

First, we compare alignment scores between correctly and incorrectly classified samples. For the fusion model, incorrect predictions are associated with lower alignment scores (mean 4.26) compared to correct predictions (mean 4.70). In contrast, the image-only model shows minimal difference between incorrect and correct cases (4.62 vs. 4.68), suggesting that alignment scores are specifically informative of the utility of generated EHRs.

Next, we analyze performance gains as a function of alignment quality. Samples are grouped into three bins: score $<4$, score $=4$, and score $=5$. The improvement of the fusion model over the image-only baseline increases monotonically with alignment score: $+2.9$, $+10.7$, and $+16.6$ percentage points, respectively.

We also examine cases where fusion improves predictions relative to the image-only model. These cases exhibit higher alignment scores (mean 4.82) compared to cases where fusion provides no benefit (mean 4.64), further supporting the role of high-quality generated modalities in enhancing multimodal learning.

Overall, these findings suggest that generation quality is a key factor in determining downstream utility. Low-quality generations do not systematically degrade performance but provide limited benefit, whereas high-quality generations contribute substantial complementary information.

## L   Cross-Evaluator Agreement Analysis

To assess the robustness of our evaluation protocol, we compare GPT-4o-based scoring with an independent evaluator, Gemini-2.5-Flash. We evaluate a subset of 120 generated samples for which both evaluators are available.

We measure both correlation and agreement metrics. The Pearson correlation between the two evaluators is 0.68, and the Spearman rank correlation is 0.76, indicating strong monotonic agreement. In terms of exact score agreement, 76% of samples receive identical scores, while 90% fall within a $\pm 1$ score difference.

These results demonstrate that the evaluation is consistent across different LLM evaluators and not overly sensitive to a specific model. Minor discrepancies typically arise in borderline cases (e.g., partial alignment vs. strong alignment) but do not affect overall trends or conclusions.

| Environmental Feature (env_feature) | Train | Val | Test | Total |
|---|---|---|---|---|
| anaerobic bioreactor | 15 | 4 | 4 | 23 |
| animal-associated habitat | 187 | 41 | 41 | 269 |
| anthropogenic contamination feature | 6 | 2 | 2 | 10 |
| banana plantation | 1 | 1 | 1 | 3 |
| bioreactor | 17 | 4 | 4 | 25 |
| bog | 8 | 2 | 2 | 12 |
| cliff | 14 | 3 | 4 | 21 |
| coast | 6 | 2 | 2 | 10 |
| coastal water body | 27 | 6 | 6 | 39 |
| coral reef | 42 | 9 | 10 | 61 |
| cropland | 6 | 2 | 2 | 10 |
| desert sand | 7 | 2 | 2 | 11 |
| forest | 18 | 4 | 5 | 27 |
| forest soil | 14 | 3 | 4 | 21 |
| glacier | 13 | 3 | 3 | 19 |
| grassland soil | 9 | 2 | 3 | 14 |
| gravel field | 6 | 2 | 2 | 10 |
| host-associated habitat | 225 | 49 | 49 | 323 |
| insecta-associated habitat | 70 | 16 | 16 | 102 |
| intermittent pond | 15 | 4 | 4 | 23 |
| kelp forest | 17 | 4 | 4 | 25 |
| lagoon | 1 | 1 | 2 | 4 |
| mangrove | 1 | 1 | 2 | 4 |
| marine benthic feature | 214 | 46 | 47 | 307 |
| marine reef | 13 | 3 | 3 | 19 |
| microbial mat | 1 | 1 | 2 | 4 |
| mine | 34 | 8 | 8 | 50 |
| montane desert | 18 | 4 | 4 | 26 |
| mud | 9 | 3 | 3 | 15 |
| ocean | 30 | 7 | 7 | 44 |
| ocean water | 12 | 3 | 3 | 18 |
| oil palm plantation | 4 | 1 | 1 | 6 |
| oil reservior | 9 | 2 | 3 | 14 |
| planted forest | 1 | 1 | 3 | 5 |
| pond bed | 43 | 10 | 10 | 63 |
| rangeland | 1 | 1 | 3 | 5 |
| research facility | 115 | 25 | 25 | 165 |
| saline marsh | 16 | 4 | 4 | 24 |
| sand | 1 | 1 | 2 | 4 |
| sediment | 9 | 2 | 3 | 14 |
| soil | 1 | 1 | 1 | 3 |
| steppe soil | 8 | 2 | 2 | 12 |
| stratum | 12 | 3 | 3 | 18 |
| stromatolite mat | 1 | 1 | 2 | 4 |
| temperate coniferous forest | 10 | 3 | 3 | 16 |
| tributary | 18 | 5 | 5 | 28 |
| volcanic feature | 42 | 9 | 9 | 60 |
| wetland | 8 | 2 | 2 | 12 |
| **Total** | **1355** | **315** | **332** | **2002** |

Table 10: EMP500 (Earth Microbiome Project Multi-omics, Study 13114) dataset statistics showing the number of samples per environmental feature across Train/Validation/Test splits.

Table 11: Comparison between the proposed fusion method and a simple concatenation baseline on the HAM10000 and EMPO test sets. Bold indicates the better result for each metric within each dataset.

| Dataset | Method | Accuracy | Macro-F1 | Macro-Precision | Macro-Recall | AUROC |
|---|---|---|---|---|---|---|
| HAM10000 | Simple Concat | 0.9499 | 0.8774 | 0.8993 | 0.8627 | 0.9826 |
| | Proposed Method | **0.9580** | **0.9140** | **0.9364** | **0.8956** | **0.9960** |
| EMPO | Simple Concat | 0.8976 | 0.6885 | 0.6911 | 0.7131 | **0.9911** |
| | Proposed Method | **0.9307** | **0.7996** | **0.8006** | **0.8342** | 0.9657 |

Table 12: Comparison with recent LLM/VLM baselines used as direct classifiers. The proposed supervised fusion model consistently outperforms general-purpose prompting-based approaches.

| Dataset | Method | Accuracy | Macro-F1 |
|---|---|---|---|
| HAM10000 | Qwen2.5-VL (direct prompting) | 0.1049 | 0.0294 |
| | GPT-4o-mini (zeroshot) | 0.1987 | 0.2656 |
| | GPT-4o-mini (Chain of Thought) | 0.3313 | 0.3198 |
| | GPT-4o-mini (fewshot) | 0.3945 | 0.3383 |
| | GPT-4o-mini (Chain of Thought with fewshot) | 0.4516 | 0.3732 |
| | BiomedCLIP (zero-shot) | 0.1392 | 0.1312 |
| | **BiomedCLIP Fusion (ours)** | **0.4914** | **0.4340** |
| EMPO | Qwen2.5-VL (direct prompting) | 0.2741 | 0.1851 |
| | GPT-4o (zeroshot) | 0.3765 | 0.2846 |
| | GPT-4o (Chain of Thought) | 0.3765 | 0.2781 |
| | GPT-4o (fewshot) | 0.6777 | 0.4815 |
| | GPT-4o (Chain of Thought with fewshot) | 0.6898 | 0.5231 |
| | BioMistral (direct prompting) | 0.0120 | 0.0005 |
| | **Fusion Model (ours)** | **0.9307** | **0.7996** |

Table 13: Expert clinical plausibility evaluation of generated EHR narratives (70 cases).

| Evaluation Criterion | Yes | No |
|---|---|---|
| Diagnosis Consistency | 67 / 70 (95.7%) | 3 / 70 |
| Internal Logical Consistency | 64 / 70 (91.4%) | 6 / 70 |

Table 14: Performance comparison for Image-to-Text direction on PAD-UFES-20.

| Metric | Image-only (Baseline) | Proposed (Ours) | Absolute Gain |
|---|---|---|---|
| Accuracy | $0.6089 \pm 0.0535$ | $0.6939 \pm 0.0413$ | +0.0850 |
| Macro-AUROC | $0.8649 \pm 0.0055$ | $0.8888 \pm 0.0089$ | +0.0239 |
| Macro-F1 | $0.4947 \pm 0.0373$ | $0.5699 \pm 0.0224$ | +0.0752 |
| Macro-Recall | $0.5044 \pm 0.0466$ | $0.6033 \pm 0.0336$ | +0.0989 |
| Weighted-F1 | $0.6205 \pm 0.0239$ | $0.6900 \pm 0.0246$ | +0.0695 |

Table 15: Performance comparison for Text-to-Image direction on PAD-UFES-20.

| Metric | Random Forest (Baseline) | Proposed (Ours) | Absolute Gain |
|---|---|---|---|
| Accuracy | $0.7436 \pm 0.0012$ | $0.6729 \pm 0.0288$ | $-0.0707$ |
| Macro-AUROC | $0.9016 \pm 0.0041$ | $0.8902 \pm 0.0066$ | $-0.0114$ |
| Macro-F1 | $0.4998 \pm 0.0007$ | $0.6400 \pm 0.0097$ | +0.1402 |
| Macro-Precision | $0.5701 \pm 0.0008$ | $0.6323 \pm 0.0155$ | +0.0622 |
| Macro-Recall | $0.4712 \pm 0.0006$ | $0.6857 \pm 0.0199$ | +0.2145 |

Table 16: Comparison between synthetic and real multimodal supervision on PAD-UFES-20 using the same multimodal fusion model.

| Setting | Accuracy | Macro-F1 |
|---|---|---|
| Metadata only | $0.3188 \pm 0.0130$ | $0.1499 \pm 0.0060$ |
| Generated image + metadata | $0.6729 \pm 0.0288$ | $0.6400 \pm 0.0097$ |
| Real image + metadata | $0.7740 \pm 0.0143$ | $0.7348 \pm 0.0157$ |

Table 17: Comparison between conventional augmentation and metadata-conditioned synthetic image augmentation on PAD-UFES-20.

| Setting | Accuracy | Macro-F1 | Macro-Recall | Macro-AUROC |
|---|---|---|---|---|
| Original image + metadata | $0.7740 \pm 0.0143$ | $0.7348 \pm 0.0157$ | $0.7266 \pm 0.0177$ | $0.9133 \pm 0.0031$ |
| + Standard augmentation | $0.7657 \pm 0.0159$ | $0.7405 \pm 0.0245$ | $0.7353 \pm 0.0351$ | $0.9129 \pm 0.0083$ |
| + Synthetic images (Ours) | $0.7884 \pm 0.0148$ | $0.7717 \pm 0.0207$ | $0.7606 \pm 0.0193$ | $0.9218 \pm 0.0117$ |

