# OpenReview forum: "Cross-Modal Generative Augmentation for Multimodal Biological Classification"
_TMLR — Accepted by TMLR_

### Review · Reviewer_RcrP · 2026-03-19

**Summary Of Contributions:**

This paper studies vision-language models scientific and biological applications. Specifically, the authors propose a cross-modal generative framework that leverages both text-to-image and image-to-text generation to enrich multimodal biological classification. The proposed framework integrates generative augmentation and multimodal alignment to mutually refine visual and textual representations, which enables the synthesis of complementary modality data that may otherwise be unavailable in biological datasets. The authors conducted empirical experiments on the HAM10000 and EMPO500 datasets and show consistent gains in accuracy and generalization.

**Audience:**

Yes

**Audience Explanation:**

This paper is on VLMs for (biological) science, and the core idea can be applied to other domains. I think main individuals in TMLR's audience will be interested in knowing the findings of this paper.

**Claims And Evidence:**

No

**Claims Explanation:**

## Claims that are supported by accurate, convincing and clear evidence

1. The proposed framework employs two independent generative augmentation pathways: image-to-text and text-to-image. These generators synthesize complementary modality data that are incorporated into a shared multimodal classifier.

    - This claim is well supported in the paper and clearly presented by using Figure 1.
    - This core idea is technically sound and I feel the practicality is high.
    - Seeing the source code, I think the implementation is nice. It would be easy to implement it from scratch.

## Claims that are *not* supported by accurate, convincing and clear evidence

1. Cross-modal generative augmentation can improve multimodal biological classification.

    - The empirical experiments on HAM10000 are limited to image-to-text, and those and EMPO500 are limited to text-to-image.
    - In this sense, the performance of the proposed method is evaluated on a single dataset for each direction.
    - I do not think the above claim on the empirical performance is fully supported on the empirical experiments.

1. Achieving 3-5% improvements over baseline models

    - This is not an accurate claim. Seeing the tables, there is no consistent 3-5% improvement over metrics and datasets.

**Requested Changes:**

1. If this paper gets accepted, please consider release the source code on GitHub and the models on Huggingface.

1. If possible, I would like to authors to add other datasets for each direction of text-to-image and image-to-texts. In the current form, the claim that "the unified bidirectional framework works because both directions mutually reinforce each other" is not sufficiently supported.

1. I feel the performance improvement is overly claimed (e.g., in Abstract). Please consider revising them.

---

> ### Author Response · Authors · 2026-04-14
>
> We sincerely thank the reviewer for the constructive feedback. We have carefully addressed the concerns regarding the empirical evidence and the phrasing of our claims. Below is a summary of our revisions and additional experiments.
>
>
>
> ### **1. Clarification of Framework & Terminology**
>
> We agree with the reviewer that the phrase *“mutually reinforce”* could imply a symmetric joint optimization, which was not our intended focus. We have revised the manuscript to clarify that the two pathways provide **complementary augmentation** to address information scarcity.
>
> * **Revisions in Abstract (Lines 7–11) & Introduction (Line 15):**
>   Replaced *“mutually reinforce/refine”* with *“provide complementary information/augmentation.”*
>
> * **Revised Contribution:**
>   We clarify that each direction targets different types of modality incompleteness:
>
>   * Image-to-Text (I2T): semantic metadata enrichment
>   * Text-to-Image (T2I): visual space expansion
>
> Importantly, our goal is **not** to demonstrate that both directions must be jointly applied or mutually reinforce each other, but rather that each direction provides complementary benefits under **different modality availability scenarios**, consistent with our direction-agnostic framework.
>
> We further emphasize that we intentionally do **not** adopt a tightly coupled or round-trip training strategy. Empirically, as reported in Appendix E, round-trip generation consistently **degraded performance due to error accumulation across modalities**, which directly motivates our design choice of independent augmentation pathways.
>
>
>
> ### **2. Additional Empirical Evidence (PAD-UFES-20 Dataset)**
>
> To address the reviewer’s concern regarding dataset coverage, we conducted additional experiments on the PAD-UFES-20 dataset. This enables evaluation of both generative directions under a unified protocol (patient-level split, 5 runs with mean ± std).
>
>
>
> #### **A. Image-to-Text Direction (Image → Generated Metadata)**
>
> This experiment demonstrates how generating synthetic clinical metadata enriches multimodal representations. Our method shows **consistent improvements across all metrics**:
>
> | Metric       | Image-only      | Ours            | Gain    |
> | ------------ | --------------- | --------------- | ------- |
> | Accuracy     | 0.6089 ± 0.0535 | 0.6939 ± 0.0413 | +0.0850 |
> | Macro-AUROC  | 0.8649 ± 0.0055 | 0.8888 ± 0.0089 | +0.0239 |
> | Macro-F1     | 0.4947 ± 0.0373 | 0.5699 ± 0.0224 | +0.0752 |
> | Macro-Recall | 0.5044 ± 0.0466 | 0.6033 ± 0.0336 | +0.0989 |
> | Weighted-F1  | 0.6205 ± 0.0239 | 0.6900 ± 0.0246 | +0.0695 |
>
>
>
> #### **B. Text-to-Image Direction (Metadata → Synthetic Image)**
>
> This experiment evaluates visual augmentation via generated images:
>
> | Metric          | Random Forest   | Ours            | Gain    |
> | --------------- | --------------- | --------------- | ------- |
> | Accuracy        | 0.7436 ± 0.0012 | 0.6729 ± 0.0288 | -0.0707 |
> | Macro-AUROC     | 0.9016 ± 0.0041 | 0.8902 ± 0.0066 | -0.0114 |
> | Macro-F1        | 0.4998 ± 0.0007 | 0.6400 ± 0.0097 | +0.1402 |
> | Macro-Precision | 0.5701 ± 0.0008 | 0.6323 ± 0.0155 | +0.0622 |
> | Macro-Recall    | 0.4712 ± 0.0006 | 0.6857 ± 0.0199 | +0.2145 |
>
> While overall accuracy decreases, we emphasize that in **imbalanced biomedical settings**, class-balanced metrics such as **macro-F1 and recall are more indicative of meaningful performance**. The substantial improvements in these metrics indicate that our method better captures **minority classes**, aligning with our findings in the main paper that gains are concentrated in underrepresented categories.
>
>
>
>
> ### **3. Adjustment of Performance Claims**
>
> We have removed the generalized claim of a *“consistent 3–5% improvement”* from the Abstract and main text.
>
> In the revised Abstract, we instead report performance more conservatively as “improvements across multiple evaluation metrics,” without suggesting a consistent or uniform gain.
>
>
>
>
>
> ### **4. Code and Model Release**
>
>
> We appreciate the suggestion to improve accessibility. While the code was included in the supplementary material, we will release the source code on GitHub and trained models on HuggingFace upon acceptance.
>
>
>
> We hope these revisions clarify the scope and contribution of our work, and we sincerely thank the reviewer for helping us improve the clarity and rigor of the paper.

---

> ### Comment · Reviewer_RcrP · 2026-04-20
> **Final comments**
>
> Thank you for the rebuttal. Additional experiments have resolved my concerns at the timing of initial review. Please carefully revise the manuscript based on the reviewers' feedback if this paper gets accepted.

---

### Review · Reviewer_ezBN · 2026-03-30

**Summary Of Contributions:**

This paper studies cross-modal generative augmentation for multimodal biological classification. The main idea is to use both image-to-text generation and text-to-image generation to create complementary modality information, then combine the original and generated modalities in a multimodal classifier. The paper’s main strengths are that it addresses an interesting and timely problem, explores two distinct biological domains, and includes several useful additional analyses such as ablations, qualitative examples, and open-weight model substitutions. The overall direction is promising. The main weaknesses are in clarity and experimental consistency. The paper does not always describe the architecture and training setup consistently across sections, which makes it hard to determine exactly what model was used in the main results. In addition, while the reported gains are encouraging, the evidence for biological validity and interpretability of the generated modalities is still somewhat limited.

**Additional Comments:**

My main concern is not the overall idea, but the lack of consistency in how the method is described. If the authors can resolve these issues and tighten the empirical presentation, the paper would be substantially stronger.

**Audience:**

Yes

**Audience Explanation:**

The topic is relevant to researchers working on multimodal learning, generative augmentation, biomedical machine learning, and scientific applications of foundation models. The paper asks a meaningful question: whether cross-modal generation can help when one modality is weak, missing, or incomplete in scientific datasets.
Some findings are useful even beyond this specific paper. In particular, the negative result on round-trip generation is interesting, since it suggests that chaining generators across modalities may introduce noise rather than improve performance. That is a valuable practical insight for others working in this area.
So although I have concerns about the current version, I believe the problem setting and empirical direction would still be of interest to part of the TMLR audience.

**Broader Impact Concerns:**

I do not see a major ethical issue that would by itself block publication, but I do think the paper should discuss overtrust risks more explicitly. In the medical setting, generated EHR-like text may appear clinically plausible even when the underlying prediction is wrong. This could make users over-trust the system’s outputs. A similar concern exists in the environmental setting, where generated images may look convincing without being fully faithful to the underlying metadata. The paper already acknowledges some of these limitations, but the broader impact discussion should state more clearly that generated cross-modal outputs should not be treated as verified explanations in high-stakes settings.

**Claims And Evidence:**

No

**Claims Explanation:**

The paper provides promising empirical results, and the reported improvements on HAM10000 and EMPO500 suggest that the proposed framework may be useful.
However, I do not think the evidence is yet clear enough to fully support the claims in its current form. My main concern is that the methodological description is inconsistent across the paper. The fusion mechanism and training procedure appear to differ between the main method section, implementation section, and appendix. Because of this, it is difficult to know exactly which architecture and optimization setup produced the reported results. I also think some of the broader claims are stronger than the current evidence supports. For example, the paper argues that the framework is model-agnostic and improves interpretability, but these claims are only demonstrated in a limited way. The interpretability evidence is mainly based on qualitative cases, automatic LLM-based evaluation, and a relatively small expert review. These are useful, but not yet fully sufficient to establish strong conclusions.
Overall, the results are encouraging, but the paper needs a clearer and more internally consistent presentation before the claims can be considered fully convincing.

**Requested Changes:**

1. **Critical:** Clarify the actual method used in the experiments. The paper currently gives inconsistent descriptions of the fusion architecture and training pipeline across different sections. The authors should clearly state the exact model, fusion mechanism, loss, optimizer, and training schedule used for each main result.

2. **Critical:** Improve reproducibility by summarizing the full experimental setup in a concise table. This would make it much easier to understand what was actually run on each dataset.

3. **Critical:** Clarify the baselines and fairness of comparison. The paper should explain more clearly which results are reproduced from prior work, which are newly implemented, and whether all methods use the same splits and preprocessing.

4. **Critical:** Better justify the claims about interpretability and biological validity. The current evidence is helpful, but still limited. At minimum, the paper should more explicitly discuss the limitations of generated modalities as explanations.

5. **Important:** Add a more detailed analysis of where the gains come from, especially for rare classes versus frequent classes.

6. **Important:** Tone down claims such as “model-agnostic” and “interpretability” unless broader evidence is added.

---

> ### Author Response · Authors · 2026-04-14
>
> We thank the reviewer for the careful reading and constructive feedback.
> We are encouraged that the empirical results are viewed as promising, and we agree that the original presentation required improved clarity and consistency.
> In the revision, we have substantially clarified the method, strengthened the experimental description, and better scoped our claims.
>
>
>
> ### **1. Method inconsistency / unclear setup**
>
> Thank you for pointing this out. We agree that the original version did not clearly specify the exact configurations used in each experiment, which may have led to confusion.
>
> In the revised manuscript, we explicitly clarify that while the framework is modular, **all reported results are obtained using fixed, dataset-specific configurations**.
>
> * In Section 3.3, we now clearly state the fusion mechanism used in each setting (**concatenation for HAM10000, gated fusion for EMPO500**).
> * We explicitly clarify that **Figure 1 illustrates a generic framework**, while the *actual fusion instantiation is dataset-specific* and fully specified in Section 3.3 and Table 6, to avoid ambiguity between conceptual and implemented designs.
> * We also explicitly connect the method description to the experimental setup by stating that the exact configurations are summarized in Section 4.2.
> * Most importantly, we introduce **Table 6 (Section 4.2)**, which provides a concise and complete summary of all configurations used in the main experiments.
>
> In addition, to eliminate any remaining ambiguity, we now provide **full training details in Appendix C**, including optimizer choices, learning rates, scheduling, and augmentation policies (e.g., two-stage training for HAM10000, EMA usage, Mixup/CutMix schedule, and calibration procedure).
>
> Finally, to further validate that the chosen design is meaningful, we include an explicit comparison against a **simple concatenation baseline (Appendix, Table 11)**, showing that the proposed fusion strategy consistently improves over naive multimodal combination.
>
> Together, these changes ensure that the architecture and training pipeline used for each result are unambiguous, internally consistent, and empirically justified.
>
>
>
> ### **2. Reproducibility**
>
> We agree that a clear and concise summary of the experimental setup is essential for reproducibility.
>
> To address this, we have added **Table 6 in Section 4.2**, which consolidates all key experimental details (fusion method, backbone, optimizer, epochs, loss, batch size, image resolution, augmentation, EMA, TTA, and calibration).
>
> We also explicitly state that:
>
> * all reported results are obtained under these fixed configurations,
> * all reproduced models are evaluated using the same data splits and evaluation pipeline,
> * and **full implementation details (including hyperparameters and training schedules) are provided in the appendix**.
>
> For example, we now specify details such as the two-stage training schedule for HAM10000, optimizer settings (AdamW / Lion), and calibration procedures (temperature scaling and bias correction), ensuring that the experiments can be faithfully reproduced.
>
> We believe this significantly improves the transparency and reproducibility of the work.
>
>
>
> ### **3. Baseline fairness**
>
> Thank you for raising this important point.
>
> In the revision, we clearly distinguish between **reported baselines and reproduced baselines**:
>
> * In Section 5.1, we explicitly state that results for InceptionV3, ResNet50, DenseNet121, and ALBEF are **reported from prior work** and included for reference only.
> * All models under “Our Models” are **fully implemented and evaluated within our pipeline** using identical data splits, preprocessing, and metrics.
> * For EMPO500 (Section 5.3), all classical baselines are evaluated within our pipeline to ensure comparability.
>
> We additionally clarify that all internally implemented baselines use the same preprocessing, splits, and evaluation protocol, ensuring a fair comparison.
>
> We believe this resolves ambiguity regarding fairness.

---

> > ### Author Response · Authors · 2026-04-14
> >
> > ### **4. Interpretability and biological validity**
> >
> > We agree that the evidence for interpretability and biological validity should be presented more cautiously.
> >
> > In the revision:
> >
> > * We explicitly clarify in **Section 5.8** that generated modalities should **not be interpreted as faithful or causal explanations**, but rather as auxiliary, human-readable signals.
> > * We further expand this limitation in the **Broader Impact section (Section 7)**, emphasizing risks such as hallucination and over-trust in high-stakes settings.
> >
> > At the same time, we strengthen the supporting evidence:
> >
> > * We include automatic consistency evaluation with **two independent evaluators (GPT-4o and Gemini)** and report agreement statistics, showing consistent alignment across evaluators (Appendix L).
> > * We provide **expert validation (Appendix I)**, where licensed physicians review generated EHR records, showing high agreement with ground-truth diagnoses and strong internal consistency.
> > * Importantly, we further analyze the relationship between **generation quality and downstream performance (Appendix K)**, showing that higher alignment scores are associated with larger fusion gains. This suggests that generated modalities are not merely plausible, but are meaningfully related to classification improvements.
> >
> > Based on these additions, we position interpretability more conservatively as **plausible auxiliary evidence rather than verified explanation**, while providing stronger empirical support for its utility.
> >
> >
> >
> > ### **5. Analysis of performance gains**
> >
> > We agree that it is important to better explain where the performance gains originate.
> >
> > In the revision (Section 5.2), we add an explicit analysis of class-wise improvements grouped by frequency.
> >
> > We observe that:
> >
> > * gains are more pronounced for underrepresented classes in HAM10000 (e.g., vasc, df),
> > * and similarly in EMPO500, improvements are concentrated in low-sample environmental categories.
> >
> > We further support this with additional analysis in Appendix K, showing that improvements increase with generation quality, particularly in low-data regimes.
> >
> > Additionally, we include **additional experiments on PAD-UFES-20 (Appendix J)** under a unified setting, which show consistent trends in improved minority-class performance (e.g., macro-F1 / recall), supporting the generality of this effect.
> >
> > Overall, these results suggest that cross-modal generation primarily improves performance in **data-scarce regimes**, rather than uniformly across all classes.
> >
> >
> >
> > ### **6. Claims and model-agnosticity**
> >
> > We agree that broader claims should be carefully scoped.
> >
> > In the revision, we refine the wording to avoid overgeneralization.
> > Specifically, we clarify that our “model-agnostic” claim refers to the **ability to substitute generative modules without modifying the downstream multimodal classifier**, rather than universal generality across all models or tasks.
> >
> > To support this, we present **generator substitution experiments in Section 5.6**, where closed-weight models are replaced with open-weight alternatives (Qwen-VL / Qwen-Image). These achieve **comparable performance, with only modest degradation in some settings (e.g., EMPO500)**, demonstrating practical flexibility while avoiding overclaiming.
> >
> > In addition, beyond the two main datasets, we include **additional experiments on PAD-UFES-20 (Appendix J)**, where both image-to-text and text-to-image directions are evaluated within a unified setting.
> > This provides further evidence that the framework supports multiple generative directions and generator choices in practice.
> >
> > We also note that, as shown in Appendix E, **round-trip (cycle) generation underperforms one-pass augmentation**, supporting our design choice of independent augmentation pathways rather than tightly coupled bidirectional training.
> >
> >
> >
> > ### **7. Broader impact and over-trust**
> >
> > We appreciate this suggestion.
> >
> > We have added a **Broader Impact section (Section 7)** that explicitly discusses:
> >
> > * bias propagation from pre-trained generators,
> > * hallucination risks,
> > * and over-trust concerns in medical and environmental applications.
> >
> > We explicitly state that generated cross-modal outputs **should not be treated as verified explanations**, especially in high-stakes settings.
> >
> >
> >
> > Overall, we believe these revisions significantly improve the clarity, consistency, and credibility of the paper, and we thank the reviewer for their valuable feedback.

---

> > > ### Comment · Reviewer_ezBN · 2026-04-25
> > > **Response to the Authors**
> > >
> > > Thank you for the detailed response and for the substantial revisions.
> > > The revised manuscript addresses many of my main concerns, especially regarding methodological clarity, reproducibility, and the scope of the claims.
> > >
> > > **Clarified experimental setup.**
> > > The added configuration table and the clearer dataset-specific descriptions make the experimental setup much easier to understand. The distinction between the conceptual framework and the actual implemented configurations is also helpful.
> > >
> > >
> > > **Improved support for interpretability and validity claims.**
> > > The added analyses on generation quality, evaluator agreement, expert validation, and class-wise gains improve the empirical support. I also appreciate the more cautious framing of interpretability and model-agnosticity.
> > >
> > > **Remaining concerns.**
> > > I also noticed a minor remaining inconsistency in the experimental description: the EMPO500 training text appears to describe AdamW, while Table 6 lists Lion as the optimizer. Please double-check this and any other remaining configuration mismatches.

---

> > > > ### Author Response · Authors · 2026-04-26
> > > >
> > > > Thank you for pointing this out. You are correct that there is a remaining textual inconsistency.
> > > >
> > > > We confirm that the EMPO500 experiments used the Lion optimizer, as reported in Table 6. The AdamW mention in the EMPO500 training description in Appendix C is an oversight.
> > > >
> > > > We will correct this in the final version and re-check the remaining configuration descriptions for consistency.

---

> > > > > ### Comment · Reviewer_ezBN · 2026-04-27
> > > > > **Response to Authors**
> > > > >
> > > > > Thank you for the clarification. Please carefully revise the manuscript based on the dicussions and reviewers' feedback.

---

### Review · Reviewer_gaFe · 2026-04-05

**Summary Of Contributions:**

The paper proposes a cross-modal generative augmentation framework using both image-to-text and text-to-image generation to improve multimodal biological classification on HAM10000 and EMPO500. It uses a visual encoder, tabular (text) encoder, and GLU fusion module to combine real and generated modalities for classification. The framework is shown to be model-agnostic via open-weight generator substitution.

**Strengths:**
Validation across two distinct biological domains; some good discussion on negative results (round-trip ablation, failure cases); clinical expert validation of generated EHRs.

**Weaknesses**
The "bidirectional" framing is misleading because each dataset uses only one generative direction, making this effectively two separate unidirectional experiments rather than a unified bidirectional framework. Several internal inconsistencies. EMPO500 gains are marginal.

**Additional Comments:**

- The abstract's "3-5% improvement" claim holds for HAM10000 but overstates EMPO500.

- Section 3.4 states 30 epochs for Stage 1, but Appendix C.2 states 40 epochs.

- Appendix H (LLM/VLM comparisons) should be moved to the main paper as it directly contextualizes the value of the proposed approach.

**Audience:**

Yes

**Audience Explanation:**

The problem of missing modalities in small biological datasets is practically important, and the discussion of failure cases makes this a valuable contribution to the community.

**Broader Impact Concerns:**

No Broader Impact Statement is present.
One is needed covering: Potential bias propagation from pre-trained generators into biological classifiers.

**Claims And Evidence:**

No

**Claims Explanation:**

The HAM10000 results are convincing across five runs with clear ablations. However, some issues weaken the evidence:
- The biggest concern is that"bidirectional" framing is never demonstrated. Each dataset uses only one generative direction, so claimed complementarity is never shown
- Even if the "bidirectional" framing is technically justified, the novelty claim is inaccurate as this concept has been explored in prior work [1, 2, 3]. While the paper briefly mentions [1], it lacks a substantive discussion on how this approach differs from existing literature. The authors should incorporate and discuss recent relevant works to better contextualize their contribution.
- Missing baselines, such as Few-shot CoT prompted GPT-4o as a classifier
- Generation quality is evaluated solely by GPT-4o while GPT-4o-mini is one of the generators, potentially a bias towards the results
-  EMPO500 gains are also marginal (<0.5% over tabular-only), further limiting the overall evidence.

[1] Back-Modality: Leveraging Modal Transformation for Data Augmentation, Li et al., NeurIPS'23

[2] Bidirectional Multimodal Knowledge Augmentation with Sparse Representation for Image-Text Retrieval, Wang et al., In 2025 IEEE International Conference on Systems, Man, and Cybernetics (SMC)

[3] BiMAC: Bidirectional Multimodal Alignment in Contrastive Learning, Zareapoor et al., AAAI'25

**Requested Changes:**

- Reframe the bidirectional claim. The paper presents itself as a unified framework leveraging both generative directions, but each dataset uses only one direction: image-to-text for HAM10000 and text-to-image for EMPO500. Both directions are never applied to the same dataset, so the complementarity of the two directions is never actually demonstrated empirically. The authors should either (a) apply both directions to at least one shared dataset and show their combined benefit, or (b) reframe the contribution as domain-adaptive unidirectional generation rather than a bidirectional framework.

- Revise the novelty claim. Need to compare the method with prior work mentioned above ([1, 2, 3]). The more defensible novelty is the biological domain application and independent (non-round-trip) deployment strategy.

- Add few-shot VLM baselines. A few-shot prompts with Chain-of-thought (CoT) GPT-4o classifier is a natural, strong baseline but absent. Zero-shot comparison alone is insufficient.

- Resolve the image-to-text input inconsistency. Section 3.1 says Gv→t takes only visual input, Figure 1 shows both image and metadata as inputs, and Section 4.1.1 reverts to image-only.  What are the actual input and output of the Gv→t?

- Address generation hallucination. No analysis exists of whether low-quality or hallucinated outputs hurt downstream performance. The failure cases in Appendix D show this might be a risk.

- Clarify Mixup in Section 3.4. "Mixup-based regularization applied with probability 0.3" does not specify what is being mixed in the multimodal setting. Is it image features, tabular embeddings, or fused representations?

- Address GPT-4o evaluator bias. Using GPT-4o to evaluate outputs generated by GPT-4o-mini risks systematic score inflation. An independent evaluator should be included.

---

> ### Author Response · Authors · 2026-04-14
>
> We thank the reviewer for their careful reading and constructive feedback. We have revised the manuscript accordingly and address each concern below.
>
>
>
> ### **1. Bidirectional framing**
>
> We thank the reviewer for this important point and agree that our original wording overstated the sense in which the framework was “bidirectional.”
>
> In the revision, we explicitly reframe the method as a **direction-agnostic, dataset-dependent augmentation framework**, rather than claiming joint bidirectional augmentation within each dataset. We revised the Introduction, Related Work, and Method sections accordingly to clarify that the two generation pathways are **independently deployable** and used depending on modality availability in a given biological setting.
>
> To more directly address the reviewer’s concern, we also added **additional experiments on PAD-UFES-20 under a unified protocol covering both directions** in the revision Appendix J. In this shared biomedical setting, the image-to-text direction improves accuracy from **0.6089 to 0.6939** and macro-F1 from **0.4947 to 0.5699**, while the text-to-image direction improves macro-F1 from **0.4998 to 0.6400** and macro-recall from **0.4712 to 0.6857** (with lower overall accuracy, reflecting improved minority-class sensitivity). These added results are intended to complement the main paper’s dataset-specific experiments and better substantiate our claim that the framework supports both generative directions in practice.
>
>
>
>
> ### **2. Novelty and relation to prior work ([1], [2], [3])**
>
>
> We thank the reviewer for highlighting this.
>
> In the revision, we expanded the Related Work section to explicitly discuss prior approaches, including Back-Modality (Li et al., 2023), as well as recent bidirectional multimodal learning methods such as Wang et al. (2025) and Zareapoor et al. (2025).
>
> We further clarified that our contribution does not lie in introducing bidirectional generation itself, but in its **independent, modular deployment for multimodal biological classification**, which differs from prior work focused on alignment, retrieval, or tightly coupled bidirectional training.
>
> We also toned down the novelty claim throughout the manuscript and now position the contribution primarily as a practical, modular augmentation strategy for small and heterogeneous biological datasets, rather than as a new form of bidirectional generation.
>
> We believe this provides a clearer and more accurate positioning of our work.
>
>
>
> ### **3. Missing few-shot VLM baselines**
>
> We thank the reviewer for this suggestion.
>
> We have added few-shot VLM baselines with chain-of-thought prompting (GPT-4o), and report these results in Table 8 and Table 12. The results show that our supervised fusion approach continues to outperform prompting-based baselines under both zero-shot and few-shot settings.
>
> Even under few-shot + CoT prompting, GPT-4o(-mini) remains substantially below the proposed supervised fusion model (e.g., 0.4516 / 0.3732 on HAM10000 and 0.6898 / 0.5231 on EMPO for accuracy / macro-F1), reinforcing the value of task-specific multimodal supervision.
>
> We believe this strengthens the empirical evaluation.
>
>
>
>
> ### **4. Image-to-text input inconsistency**
>
>
> We thank the reviewer for pointing this out.
>
> We have clarified that, in HAM10000, the image-to-text generator is conditioned on the dermoscopic image together with structured metadata (age, sex, localization), and its output is a structured EHR-like record that is subsequently encoded by the tabular branch.
>
> This resolves the discrepancy between the text and the figure.

---

> > ### Author Response · Authors · 2026-04-14
> >
> > ### **5. Generation hallucination**
> >
> > We agree that the effect of low-quality or hallucinated generations on downstream performance should be analyzed more explicitly.
> >
> > In the revision, we added a detailed sample-level analysis linking generation quality to downstream outcomes on HAM10000. Using alignment scores assigned to generated EHRs, we find that incorrectly classified samples in the fusion model have lower average alignment scores than correctly classified ones (4.26 vs. 4.70). Moreover, the fusion model’s improvement over the image-only baseline increases monotonically with alignment quality (+2.9, +10.7, and +16.6 percentage points across increasing alignment bins). These results suggest that low-quality generations provide limited benefit, whereas high-quality generations contribute substantial complementary information.
> >
> > In our analysis, lower-quality generations were associated with reduced benefit rather than clear systematic degradation at the aggregate level. However, as illustrated in our qualitative failure cases, hallucinated or misaligned generations can still negatively affect individual predictions, and we highlight this as an important limitation and direction for future work.
> >
> >
> > ### **6. Mixup ambiguity**
> >
> >
> > We thank the reviewer for pointing out this ambiguity.
> >
> > We have revised Section 3.4 to explicitly describe the augmentation strategy. In HAM10000, Mixup/CutMix is applied only to the image tensor and labels during the first 10 epochs of Stage 1, while EHR metadata remains unchanged. In EMPO500, we use image-only interpolation regularization with probability 0.3, without mixing tabular features or labels.
> >
> > We believe this clarification resolves the issue.
> >
> >
> >
> > ### **7. GPT-4o evaluator bias**
> >
> > We agree that relying on GPT-4o alone as an evaluator could introduce bias, especially when GPT-4o-mini is used as one of the generators.
> >
> > To address this, we added an independent cross-evaluator analysis using Gemini-2.5-Flash on a subset of 120 samples. The two evaluators show consistent agreement (Pearson 0.68, Spearman 0.76, 76% exact match), suggesting that the observed trends are not driven by a single evaluator. In addition, for the HAM10000 image-to-text setting, we report a small-scale expert validation by two licensed physicians over 70 generated EHRs, with 95.7% diagnosis consistency and 91.4% internal logical consistency, providing an orthogonal check beyond LLM-based scoring.
> >
> > For additional context, the generated EHRs and environmental images achieve strong average alignment scores of 4.40/5 and 4.56/5, respectively, indicating that the evaluated samples are not dominated by obviously poor generations.
> >
> >
> >
> >
> > ### **8. Broader Impact**
> >
> >
> >
> > We thank the reviewer for this suggestion.
> >
> > We have added a Broader Impact section discussing potential risks, including bias propagation from pre-trained generators, hallucinated outputs, and implications for real-world biological or clinical deployment.
> >
> >
> >
> > ### **9. EMPO500 gains are marginal**
> >
> > We agree that the gains on EMPO500 are modest and have revised the abstract and discussion to avoid overstatement. We now present EMPO500 primarily as evidence of transferability across domains, rather than as a setting with large absolute gains. We attribute the modest effect size to the limited visual signal and metadata-driven nature of the dataset, and clarified this interpretation in the revised manuscript.
> >
> >
> >
> > ### **10. Additional comments**
> >
> > - We revised the abstract to avoid overstatement of improvements on EMPO500.
> > - We resolved inconsistencies in training epochs between Section 3.4 and Appendix C.2.
> > - We included the key results from Appendix H (LLM/VLM comparisons) in the main paper to better contextualize the contribution.
> >
> >
> > We thank the reviewer again for the valuable feedback, which has significantly improved the clarity and positioning of our work.

---

> ### Comment · Reviewer_gaFe · 2026-04-15
> **Thank you for the rebuttal & some follow-up questions**
>
> Thank you to the authors for the thorough rebuttal. The response has largely addressed my questions and improved my understanding of the work.
> I have some follow-ups:
>
> **1. PAD-UFES-20 & Augmentation Logic**
> The addition of the PAD-UFES-20 dataset in Appendix J is helpful, but more detail is needed. Please provide a formal citation and describe the specific modalities it contains. Regarding the text-to-image direction: since the dataset already includes images, could you clarify the implementation? Specifically, are the synthetic and original images combined via feature-level fusion, or are they treated as separate augmented samples in a complementary ensemble? How difference between the original image and its generation? Similar questions on the text with image to text generation.
>
> **2. Addressing Generation Hallucination**
> The revision contains Section K, which analyzes generation quality, but it lacks a specific analysis for "biological hallucination." For example, a model might generate a visually plausible image that is not well supported by the underlying clinical metadata. How do you quantify the risk of medically incorrect features? Furthermore, how does the framework handle variance when different models (e.g., GPT vs. Qwen-VL) generate different but "sensible" outputs for the same prompt?
>
> **3. Model-Agnostic Verification (Table 7)**
> Regarding the results in Table 7, I would like more insight into the "source" of the performance gain. Is the improvement driven primarily by the raw quality/resolution of the generated assets, or is the semantic alignment with the metadata the more critical factor?

---

> > ### Author Response · Authors · 2026-04-15
> >
> > We thank the reviewer for the thoughtful follow-up questions. We provide clarifications below.
> >
> >
> >
> > ### **1. PAD-UFES-20 & Augmentation Logic**
> >
> > We agree that our description of PAD-UFES-20 in the appendix lacked sufficient detail.
> >
> > PAD-UFES-20 is a dermatological dataset[1] consisting of dermoscopic images paired with clinical metadata (e.g., age, sex, lesion location, diagnostic labels), making it suitable for evaluating both image-to-text and text-to-image directions within a unified setting, introduced for analysis purposes and complementing the dataset-dependent design in the main experiments.
> >
> > In our experiments, the two generative directions are applied independently under a shared protocol:
> >
> > * **Image-to-text** uses dermoscopic images with structured metadata fields (e.g., age, sex, lesion location) to generate structured EHR-like descriptions, which are encoded as auxiliary tabular inputs.
> > * **Text-to-image** uses metadata to generate synthetic images, which are introduced as additional training samples.
> >
> > Importantly, in the text-to-image direction, generated images are not combined with original images at the feature level. Instead, they are used as additional training samples paired with metadata. More generally, each generative direction augments a complementary modality (i.e., generated EHR for image-to-text, and generated images for text-to-image), rather than mixing real and generated data within the same modality.
> >
> > The generated images are not intended to replicate the original instance, but rather to provide **plausible, metadata-consistent variations** that expand the visual feature space.
> >
> >
> >
> > ### **2. Biological Hallucination and Inter-Model Variance**
> >
> > We agree that “biological hallucination” deserves more precise characterization.
> >
> > In the current work, we do not explicitly quantify hallucination as inconsistency between generated outputs and ground-truth biological attributes. Instead, we use **alignment scores as a proxy for semantic consistency**, where generated outputs are evaluated against their conditioning inputs (Section 5.7) .
> >
> > While this does not fully capture medically incorrect but plausible generations, our analysis (Appendix K) shows that **lower alignment scores are associated with reduced downstream gains**, indicating that misaligned generations contribute less effectively to the model.
> >
> > Regarding variance across generators (e.g., GPT vs. Qwen-VL), we did not explicitly measure output diversity for the same input. However, as shown in Table 7, substituting different generators results in **comparable downstream performance**, suggesting that the framework is robust to variation across plausible generations, as long as semantic consistency is preserved.
> >
> > We agree that more fine-grained analysis of hallucination and inter-generator variance is an important direction for future work.
> >
> >
> >
> > ### **3. Source of Performance Gains (Quality vs. Alignment)**
> >
> > We thank the reviewer for this important question.
> >
> > While we did not explicitly disentangle perceptual quality from semantic alignment, our results suggest that **semantic alignment is the key driver of performance gains**.
> >
> > Specifically, our generation-quality analysis (Appendix K) shows that samples with higher alignment scores consistently yield larger improvements in the fusion model . In contrast, visually plausible but weakly aligned samples provide limited benefit.
> >
> > This interpretation is further supported by Table 7, where replacing closed-weight generators with open-weight alternatives leads to only minor performance differences, despite potential variation in visual quality. This indicates that **strict perceptual fidelity is not the dominant factor**, as long as the generated modality remains semantically consistent with the conditioning metadata.
> >
> >
> >
> > We appreciate the reviewer’s insightful questions, which helped us clarify the augmentation mechanism, the role of generation quality, and the source of performance improvements. These clarifications will be incorporated into the manuscript.
> >
> > ---
> >
> > [1] Pacheco AGC, Lima GR, Salomão AS, Krohling B, Biral IP, de Angelo GG, Alves FCR Jr, Esgario JGM, Simora AC, Castro PBC, Rodrigues FB, Frasson PHL, Krohling RA, Knidel H, Santos MCS, do Espírito Santo RB, Macedo TLSG, Canuto TRP, de Barros LFS. PAD-UFES-20: A skin lesion dataset composed of patient data and clinical images collected from smartphones. Data Brief. 2020 Aug 25;32:106221. doi: 10.1016/j.dib.2020.106221. PMID: 32939378; PMCID: PMC7479321.

---

> ### Comment · Reviewer_gaFe · 2026-04-18
> **Thank you and Follow-up on PAD-UFES-20**
>
> Thanks for the detailed responses!
>
> I have some follow-ups on the PAD-UFES-20's setup:
>
> 1/  If the authors are using synthetic images as extra training samples, it would be fairer to compare them against other augmentation methods.
>
> 2/ The core contribution of this paper is using pretrained generative models to synthesize missing modalities (e.g., generating images from text) to improve multimodal classification. Thus, it is interesting to see the gap between the synthetic augmentation and the ground-truth multimodal performance, specifically, apply the proposed method to the PAD-UFES-20 text-only version (i.e., use text to generate images, without original images), then compare it against the PAD-UFES-20 full dual-modality version (real text + real images, trained on the same model). Comparing these two would give us a much clearer picture of how well the generative framework approximates real-world data.

---

> > ### Author Response · Authors · 2026-04-19
> >
> > We thank the reviewer for the insightful questions and suggestions.
> >
> > ### **1. Comparison with standard augmentation**
> >
> > We agree that comparing synthetic images against standard augmentation provides a fairer assessment.
> >
> > We therefore conducted an additional control experiment on PAD-UFES-20 (text-to-image setting), comparing:
> > (i) original image + metadata,
> > (ii) original image + metadata with standard image augmentations (RandomResizedCrop, RandAugment, RandomErasing), and
> > (iii) original image + metadata with synthetic images generated from metadata.
> >
> > | Setting                   | Accuracy            | Macro-F1            | Macro-Recall        | Macro-AUROC         |
> > | ------------------------- | ------------------- | ------------------- | ------------------- | ------------------- |
> > | Original Image + Metadata | 0.7740 ± 0.0143     | 0.7348 ± 0.0157     | 0.7266 ± 0.0177     | 0.9133 ± 0.0031     |
> > | + Standard Augmentation   | 0.7657 ± 0.0159     | 0.7405 ± 0.0245     | 0.7353 ± 0.0351     | 0.9129 ± 0.0083     |
> > | + Synthetic Images (Ours) | **0.7884 ± 0.0148** | **0.7717 ± 0.0207** | **0.7606 ± 0.0193** | **0.9218 ± 0.0117** |
> >
> > Synthetic images show improved performance over both the original and standard augmentation baselines in this setting.
> >
> > ---
> >
> > ### **2. Gap to real multimodal performance**
> >
> > We agree that comparing against full multimodal supervision is informative.
> >
> > In PAD-UFES-20 (Appendix J), we evaluate the text-only setting where images are synthesized from metadata. While this improves the tabular-only baseline (e.g., +0.1402 macro-F1, +0.2145 recall), it still underperforms the full dual-modality setting with real images.
> >
> > Specifically, the synthetic-only setting (Appendix J) achieves lower performance than the real image + metadata setting reported in the table above, highlighting the gap between synthetic modality reconstruction and true multimodal supervision with real data.

---

> ### Comment · Reviewer_gaFe · 2026-04-19
> **Thank you for the answers**
>
> Thank you to the authors for the quick responses.
> Table 15 in Appendix J doesn't seem to answer the question on "Gap to real multimodal performance". First, it doesn't include a full dual-modality result using the original text and image. Second, it would be fairer to use the same model rather than comparing it with Random Forest.

---

> > ### Author Response · Authors · 2026-04-19
> >
> > We thank the reviewer for the clarification. We agree that a fair comparison should be conducted using the same model.
> >
> > In our current results, Table 15 (Appendix J) focuses on the text-only setting with generated images and reports performance relative to a tabular baseline (Random Forest). However, the corresponding full dual-modality result using the same model (real image + metadata) is reported in our previous response above.
> >
> > For clarity, we explicitly restate the same-model comparison here:
> >
> > * **Metadata only (same model):** 0.3188 ± 0.130 accuracy, 0.1499 ± 0.060 macro-F1
> > * **Generated image + metadata (same model):** 0.6729 ± 0.0288 accuracy, 0.6400 ± 0.0097 macro-F1
> > * **Real image + metadata (same model):** 0.7740 ± 0.0143 accuracy, 0.7348 ± 0.0157 macro-F1
> >
> > This makes the comparison consistent under the same model. We observe that synthetic images provide substantial improvements over the metadata-only setting, while still not fully matching the performance of true multimodal supervision using real images.
> >
> > We will revise the manuscript to present this comparison more explicitly.

---

> > > ### Comment · Reviewer_gaFe · 2026-04-19
> > > **Acknowledgement of additional results**
> > >
> > > Thanks for the quick update and for providing those results. This clears things up for me, and I'll be sure to factor these into my final recommendation. I really appreciate the helpful discussion.

---

### Decision · Action_Editor_YDMt · 2026-04-28

**Recommendation:** Accept with minor revision

**Additional Comments:**

After the initial rebuttal there was additional discussion to clarify some points that were not sufficiently addressed in the rebuttal.  These changes should be incorporated into the camera ready and will be validated.  This includes author's discussion on:

1. PAD-UFES-20 clarification
2. Comparison with standard augmentation
3. Clarification on the optimizer used for EMPO500
4. Source of Performance Gains (Quality vs. Alignment)

**Audience:**

Yes

**Audience Explanation:**

This work primary focuses on the application of known techniques to a new domain, namely classifying biological images.  These types of datasets have unique challenges that means many methods developed for natural images do not transfer.  As such, it is valuable to those with interest in this topic to have a better understanding of cases where this transfer does work well.

**Claims And Evidence:**

Yes

**Claims Explanation:**

Two of the three reviewers were fully satisfied with the paper's evidence post-rebuttal, which greatly improved over the submitted version.  One reviewer disagreed, but the AE finds that most of these criticisms are addressed in the rebuttal and could be satisfied with a minor revision of the paper that incorporates results from the rebuttal.  As such, the paper is recommended for acceptance as long as the authors can make these changes in their camera ready.